# Decision-Focused Learning via Tangent-Space Projection of Prediction Error

**Junhyeong Lee** [1]  **Sangjin Jin** [1]  **Yongjae Lee** [1 †]

## Abstract

Decision-Focused Learning (DFL) trains predictors to improve downstream decision quality, but computing regret gradients typically requires differentiating through solvers or relying on surrogate losses, which can be computationally expensive or deviate from the true objective. We show that, under standard regularity with locally stable active constraints, the regret gradient admits a closed-form geometric characterization, equivalent to the prediction error projected onto the tangent space of active constraints, scaled by local curvature. This reveals that regret gradients can be obtained by filtering decision-irrelevant components from the MSE gradient, providing a simpler and more direct alternative to existing approaches. We propose **PEAR** (**P**rojected **E**rror **A**s **R**egret-gradient), which computes regret gradients via a reduced linear system over active constraints, avoiding differentiation through solver iterations or additional optimization solves. Experiments on LP benchmarks and a real-world QP task show that PEAR achieves the best decision quality among all baselines while being the most computationally efficient, with gains that persist under constraint shifts. Code is available at [GitHub].

## 1. Introduction

Machine learning is increasingly deployed in decision-making, where a predictive model estimates parameters of an optimization problem and a downstream solver computes the final decision. In these predict-then-optimize settings, reducing the prediction error alone does not necessarily yield good decisions. This has motivated Decision-Focused Learning (DFL) (Donti et al., 2017; Elmachtoub & Grigas, 2022), which trains predictors to improve decision quality,

typically measured by regret, the suboptimality incurred by acting on predicted rather than true parameters.

Importantly, prediction accuracy and decision quality are not inherently at odds. In the linear-cost setting, minimizing mean squared error can be consistent with reducing expected regret (Elmachtoub & Grigas, 2022). This raises a natural question about the structural relationship between prediction error and regret gradients, and how this relationship can inform training. However, this connection remains underexplored. Most prior work has instead focused on procedural questions, namely how to obtain regret gradients.

While such methods are valuable for enabling end-to-end learning, existing approaches have notable limitations. Among sensitivity-based differentiable optimization layers, general-purpose methods can trade off efficiency and robustness for broad applicability, and certain constructions require symmetrizing the KKT system, which can lead to ill-conditioned linear solves (Magoon et al., 2026; Bambade et al., 2024). A complementary line of work uses surrogate objectives or perturbed estimators to approximate regret gradients. These methods can provide informative learning signals, but they may deviate from the true regret objective, and many are tailored to linear-cost problems, which limits their generality beyond linear objectives (Mandi et al., 2024; Schutte et al., 2024).

We propose **PEAR** (**P**rojected **E**rror **A**s **R**egret-gradient), a simple projection-based method for computing regret gradients. In contrast to prior work that mainly focuses on how to differentiate through an optimizer or design proxy objectives, we study the *structure* of regret gradients and their connection to prediction error. Under standard regularity and locally fixed active constraints, we show that the regret gradient admits an explicit geometric form. It is given by projecting the prediction error onto the tangent space of the active constraints under the local curvature metric. This result shows that the regret gradient can be obtained from the MSE gradient by filtering out decision-irrelevant components of the prediction error.

Guided by this characterization, PEAR computes regret gradients without differentiating through solver iterations and without relying on surrogate losses. Instead, it evaluates the required projection by solving a compact reduced linear system induced by the active constraints, e.g., via a Schur-

---

†Corresponding author. [1]Department of Industrial Engineering, Ulsan National Institute of Science and Technology, Ulsan, South Korea. Correspondence to: Yongjae Lee <yongjaelee@unist.ac.kr>.

*Proceedings of the 43rd International Conference on Machine Learning*, Seoul, South Korea. PMLR 306, 2026. Copyright 2026 by the author(s).

complement computation. This avoids backward overhead and improves numerical stability compared to differentiable layers, while remaining faithful to the true regret objective rather than a proxy. PEAR applies to optimization problems with linear or quadratic objectives.

**Our contributions are as follows:**

1. **PEAR: a simple projection for regret gradients.** We establish an explicit geometric relationship between prediction error and regret gradients in Decision-Focused Learning. This interpretation reveals which error components influence decisions, and **PEAR** filters out decision-irrelevant directions to improve DFL performance.

2. **Efficient and stable computation.** PEAR evaluates the projection by solving a compact reduced linear system induced by the active constraints, reducing backward overhead and improving numerical stability compared to differentiable optimization layers. Compared to LP baselines, PEAR provides a deterministic regret gradient without requiring additional solver calls or perturbation-based estimation.

3. **Strong empirical performance.** On synthetic LP benchmarks and a real-world QP task, PEAR improves regret and training efficiency over other baselines in most settings, and remains strong under noise and constraint perturbations.

## 2. Related Work

**Decision-focused learning.** Decision-Focused Learning (DFL) trains predictors to improve downstream decision quality, typically measured by regret, in predict-then-optimize pipelines (Donti et al., 2017; Wilder et al., 2019; Elmachtoub & Grigas, 2022). DFL has been studied in a range of applications including healthcare, portfolio optimization, and other optimization-driven decision systems (Verma et al., 2023; Costa & Iyengar, 2023; Lee et al., 2025; Hwang et al., 2025; Ellinas et al., 2024). Recent surveys summarize major methodological themes such as regularization-based approaches, convex surrogate losses, and perturbation-based gradient estimators (Mandi et al., 2024; Sadana et al., 2025). A recurring practical challenge is to obtain learning signals that are well aligned with regret while keeping training efficient, especially when training repeatedly interacts with a downstream optimization (Mandi et al., 2020; Mulamba et al., 2020; Berden et al., 2026).

**Differentiable optimization layers.** A common approach to end-to-end learning is to treat the downstream optimization as a differentiable layer. Early works relied on unrolling iterative optimization steps to compute gradients via the chain rule (Domke, 2012; Amos et al., 2017), while

OptNet (Amos & Kolter, 2017) introduced implicit differentiation through KKT conditions for quadratic programs. Subsequent work generalized this idea to broader classes of convex programs (Agrawal et al., 2019a;b; Gould et al., 2021; Blondel et al., 2022). While these methods provide principled gradients, in practice the backward pass depends heavily on the solver implementation. For instance, OptNet requires symmetrized linear systems that can become ill-conditioned (Bambade et al., 2024), while CVXPYLayers (Agrawal et al., 2019a) routes QPs through cone reformulations, restricting access to specialized solvers and degrading scalability on large sparse instances (Magoon et al., 2026). These issues make training expensive and limit solver choices (Blondel et al., 2022), motivating recent work on decoupling the forward and backward passes for solver-agnostic sensitivity analysis (Magoon et al., 2026).

**Alternative approaches for regret minimization.** Many real-world optimization problems are inherently discrete, leading to piecewise-constant solution maps that yield zero or undefined gradients (Mandi et al., 2022). To address uninformative gradients, DFL has largely focused on surrogate objectives and gradient approximations. For linear optimization, SPO+ (Elmachtoub & Grigas, 2022) is a convex surrogate that upper bounds regret and can be optimized efficiently. A related strategy smooths the optimization problem itself, for example by adding quadratic regularization or using continuous relaxations to obtain meaningful gradients (Wilder et al., 2019; Mandi & Guns, 2020). Alternatively, perturbation-based methods keep the original solver but add random perturbations and use the resulting changes in the solution to build informative gradient estimates (Berthet et al., 2020; Pogančić et al., 2019; Niepert et al., 2021). These approaches can be practical and scalable, but they do not generally yield the true regret gradient and are often restricted to linear objectives, making generalization to broader problem classes difficult (Mandi et al., 2024; Schutte et al., 2024).

**Sensitivity analysis and active set reduction.** Our analysis builds on classical sensitivity theory for constrained optimization, which characterizes how solutions change under small perturbations (Fiacco, 1976; 1983; Robinson, 1980; Bonnans & Shapiro, 2013). Under standard regularity conditions, the active constraints remain locally stable and an inequality-constrained problem reduces to an equality-constrained system over the active set (Kyparisis, 1990; Dontchev & Rockafellar, 2009). This active set reduction yields efficient computations, where solution derivatives can be obtained from reduced linear systems defined by the active constraints (Bemporad et al., 2002). These principles enable differentiable optimization to exploit active constraint reduction across arbitrary solvers, computing gradients from reduced linear systems (Magoon et al., 2026).

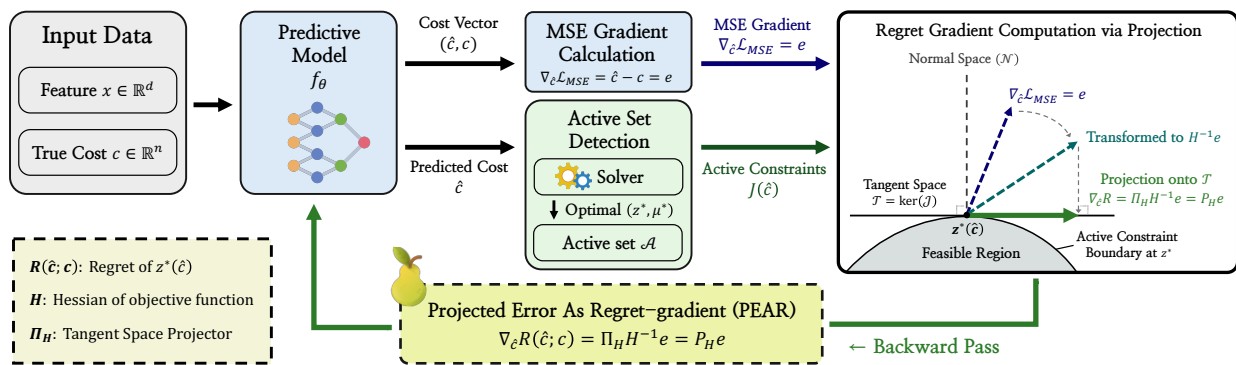

*Figure 1.* **The PEAR framework.** PEAR provides a direct pipeline for transforming prediction error into a regret gradient via a projection onto the local decision geometry. After identifying the active constraints, PEAR rescales the error by local curvature and projects it onto the feasible tangent space, filtering out decision-irrelevant components.

# 3. Regret Gradient via Tangent-Space Projection

This section establishes our main theoretical result. Under standard regularity conditions, the regret gradient admits a closed-form geometric characterization as the prediction error projected onto the tangent space of active constraints, scaled by local curvature. This structure reveals not only how to compute regret gradients efficiently by filtering decision-irrelevant components, but also why and what to correct for improving decisions.

## 3.1. Problem Setup

We consider a predict-then-optimize pipeline where contextual features $x \in \mathbb{R}^d$ inform predictions about unknown cost parameters. A predictive model $f_\theta : \mathbb{R}^d \to \mathbb{R}^n$ outputs $\hat{c} = f_\theta(x)$ as an estimate of the true cost vector $c \in \mathbb{R}^n$. Given $\hat{c}$, the downstream decision is obtained by solving the convex optimization problem

$$
\begin{aligned}
z^*(\hat{c}) \in \arg\min_{z \in \mathbb{R}^n} \quad & \phi(z) + \hat{c}^\top z \\
\text{subject to} \quad & Az = b, \\
& Gz \leq h,
\end{aligned}
\tag{1}
$$

where $\phi : \mathbb{R}^n \to \mathbb{R}$ is a known convex base objective, $A \in \mathbb{R}^{p \times n}$ defines equality constraints, and $G \in \mathbb{R}^{m \times n}$ defines inequality constraints.

Decision-Focused Learning minimizes the regret of the decision computed from $\hat{c}$ under the true cost $c$:

$$
\begin{aligned}
\mathcal{R}(\hat{c}; c) = & \left[ \phi(z^*(\hat{c})) + c^\top z^*(\hat{c}) \right] \\
& - \left[ \phi(z^*(c)) + c^\top z^*(c) \right].
\end{aligned}
\tag{2}
$$

Gradient-based training requires $\nabla_\theta \mathcal{R}$. By the chain rule,

$$
\frac{d\mathcal{R}}{d\theta} = \frac{\partial \mathcal{R}}{\partial z^*} \cdot \frac{\partial z^*}{\partial \hat{c}} \cdot \frac{\partial \hat{c}}{\partial \theta},
\tag{3}
$$

where the main challenge is the solution sensitivity $\partial z^*/\partial \hat{c}$, since it requires differentiating through an $\arg\min$ operator (Wilder et al., 2019).

Since the ground-truth cost term $c^\top z^*(c)$ in (2) is independent of $\hat{c}$, the regret gradient with respect to $\hat{c}$ becomes

$$
\nabla_{\hat{c}} \mathcal{R} = \left( \frac{\partial z^*}{\partial \hat{c}} \right)^\top \left[ \nabla_{z^*} \phi(z^*) + c \right].
\tag{4}
$$

## 3.2. Active Set and Local Regularity

The difficulty in differentiating $z^*(\hat{c})$ stems from inequality constraints. The mapping $\hat{c} \mapsto z^*(\hat{c})$ can be nonsmooth when the set of binding constraints changes. To recover local differentiability, we restrict attention to neighborhoods where the active set remains invariant. Within such a region, the inequality-constrained problem reduces to an equality-constrained one, in which inactive constraints are automatically satisfied and only the active constraints matter. Let $\mathcal{A}(\hat{c}) = \{i : (Gz^*(\hat{c}) - h)_i = 0\}$ denote the active set at $z^*(\hat{c})$ and define the stacked active constraint matrix

$$
J(\hat{c}) = \begin{bmatrix} A \\ G_{\mathcal{A}(\hat{c})} \end{bmatrix} \in \mathbb{R}^{k \times n}, \qquad k = p + |\mathcal{A}(\hat{c})|.
\tag{5}
$$

On any neighborhood where $\mathcal{A}(\hat{c})$ remains unchanged, the inequality-constrained problem behaves like an equality-constrained problem with constraint Jacobian $J$. To ensure the solution map remains smooth within this region, we require the following regularity conditions from classical sensitivity analysis (Fiacco, 1983).

**Assumption 3.1** (Strict Convexity). The Hessian $H = \nabla^2 \phi(z^*)$ is positive definite.

**Assumption 3.2** (Linear Independence Constraint Qualification). The active constraint Jacobian has full row rank, $\mathrm{rank}(J) = k$.

**Assumption 3.3** (Strict Complementary Slackness). For each active inequality $i \in \mathcal{A}$, the corresponding dual variable satisfies $\mu_i^* > 0$; for inactive constraints, $\mu_i^* = 0$.

Under these assumptions, the solution map is locally unique and continuous, and the active set remains invariant under sufficiently small perturbations of $\hat{c}$ (Robinson, 1980; Fiacco, 1983; Bonnans & Shapiro, 2013). A formal statement is provided in Appendix A.1.

### 3.3. Sensitivity via a Reduced KKT System

Fix $\hat{c}$ and consider perturbations $d\hat{c}$ that are small enough so that the active set does not change. At the optimal solution, the KKT conditions are given by

$$\nabla\phi(z^*) + \hat{c} + A^\top\lambda^* + G^\top\mu^* = 0,$$
$$Az^* = b, \quad Gz^* \leq h, \quad \mu^* \geq 0, \quad (6)$$
$$\mathrm{diag}(\mu^*)(Gz^* - h) = 0.$$

Under Assumptions 3.1–3.3, inactive inequalities remain strictly feasible under sufficiently small perturbations that preserve the active set. Thus the problem locally reduces to an equality-constrained system whose KKT conditions become

$$\nabla\phi(z^*) + \hat{c} + J^\top y^* = 0, \qquad Jz^* = \tilde{b}, \quad (7)$$

where $J$ is the stacked active constraint Jacobian, $y^* \in \mathbb{R}^k$ collects dual variables for active constraints and $\tilde{b} = [b; h_{\mathcal{A}}]$. We can view the reduced KKT system (7) as implicitly defining $z^*$ and $y^*$ as functions of $\hat{c}$. Define the implicit function

$$F(z, y, \hat{c}) = \begin{bmatrix} \nabla\phi(z) + \hat{c} + J^\top y \\ Jz - \tilde{b} \end{bmatrix} = 0. \quad (8)$$

By the implicit function theorem (Dontchev & Rockafellar, 2009), differentiating $F$ with respect to $\hat{c}$ yields an explicit linear system for the perturbations $dz$ and $dy$

$$\begin{bmatrix} H & J^\top \\ J & 0 \end{bmatrix} \begin{bmatrix} dz \\ dy \end{bmatrix} = \begin{bmatrix} -d\hat{c} \\ 0 \end{bmatrix}. \quad (9)$$

where $H = \nabla^2\phi(z^*)$ is the Hessian of the objective.

Since the active set is fixed, only the active constraints appear in the system, making this explicit formula much simpler than working with the full KKT conditions where many dual variables would be zero.

Under Assumptions 3.1–3.2, the KKT matrix in (9) is invertible (Boyd & Vandenberghe, 2004). Eliminating the dual variable $dy$ via the Schur complement gives

$$dz = -P_H\, d\hat{c}, \qquad \text{equivalently} \qquad \frac{\partial z^*}{\partial \hat{c}} = -P_H, \quad (10)$$

where the sensitivity operator is

$$P_H = H^{-1} - H^{-1}J^\top(JH^{-1}J^\top)^{-1}JH^{-1}. \quad (11)$$

The derivation of (10)–(11) is standard in sensitivity analysis. For completeness, we include it in Appendix A.2.

The operator $P_H$ enforces feasibility at first order. Indeed, substituting $dz = -P_H d\hat{c}$ into the linearized constraints yields $Jdz = 0$. Therefore, the first-order solution perturbations lie in the feasible tangent space $\mathcal{T} := \ker(J)$. This is made explicit by the projector identity below.

**Proposition 3.4** (Projection Structure). *Under Assumptions 3.1–3.2, define*

$$\Pi_H = I - H^{-1}J^\top(JH^{-1}J^\top)^{-1}J. \quad (12)$$

*Then $\Pi_H$ is an $H$-orthogonal projector onto $\ker(J)$ and*

$$P_H = \Pi_H H^{-1}, \qquad JP_H = 0, \qquad P_H J^\top = 0. \quad (13)$$

Proposition 3.4 shows that $P_H$ can be read as curvature scaling followed by projection onto feasible directions. The proof is given in Appendix B.1.

### 3.4. Regret Gradient and the Projected Prediction Error

We now connect the sensitivity operator to the regret gradient. Let $e = \hat{c} - c$ denote the prediction error. Under Assumptions 3.1–3.3, the regret is differentiable with respect to $\hat{c}$ on any neighborhood where the active set is fixed. This allows the regret gradient in (4) to be simplified in closed form via the sensitivity operator $P_H$ defined in (11).

**Theorem 3.5** (Regret Gradient via Tangent-Space Projection). *Under Assumptions 3.1–3.3, the regret gradient admits the closed form*

$$\nabla_{\hat{c}}\mathcal{R}(\hat{c}; c) = P_H(\hat{c} - c) = P_H e,$$

*where $P_H$ is defined in (11).*

Theorem 3.5 implies that regret descent replaces the MSE gradient $e$ with a projected error $P_H e$ induced by the local active constraints and curvature. The proof combines the chain rule with the sensitivity relation (10) and the projection property in Proposition 3.4. A complete proof appears in Appendix B.2.

Since $P_H$ projects onto $\mathcal{T} = \ker(J)$, it removes all normal components, leading to two key properties. First, the learning signal filters out normal directions. Second, it remains invariant to normal perturbations whenever the active set is fixed. These properties are formalized as follows.

**Corollary 3.6** (Normal Filtering and Normal Invariance). *Let $\mathcal{N} = \mathrm{range}(J^\top)$ denote the normal space to the tangent space $\mathcal{T}$. For any $\delta_N \in \mathcal{N}$,*

$$P_H \delta_N = 0. \quad (14)$$

*If the active set remains unchanged, then for all $\alpha \in \mathbb{R}$,*

$$\nabla_{\hat{c}}\mathcal{R}(\hat{c} + \alpha\delta_N; c) = \nabla_{\hat{c}}\mathcal{R}(\hat{c}; c). \quad (15)$$

A proof is given in Appendix B.3. Corollary 3.6 exposes the structural advantage of DFL. Prediction errors orthogonal to the feasible region do not affect the downstream decision, so DFL naturally suppresses such errors.

As shown in Figure 1, $H^{-1}$ rescales the prediction error $e$ according to the local curvature of the objective $\phi$. In directions where the objective is flatter, smaller eigenvalues of $H$ amplify the corresponding error components through $H^{-1}e$, while directions with steep curvature are correspondingly suppressed (Nocedal & Wright, 2006). This curvature-weighted error reveals which cost perturbations most affect the solution across different geometric directions.

The tangent projector then filters this rescaled error by projecting onto the feasible tangent space $\mathcal{T} = \ker(J)$, removing all components orthogonal to this space. The complete regret gradient is thus

$$\nabla_{\hat{c}}\mathcal{R}(\hat{c};c) = \underbrace{\left(I - H^{-1}J^{\top}(JH^{-1}J^{\top})^{-1}J\right)}_{\text{tangent projector}} \underbrace{\left(H^{-1}\right)}_{\substack{\text{curvature} \\ \text{rescaler}}} e,$$

which first rescales by local curvature, then projects onto the tangent space of feasible directions. This geometric decomposition is also characterized in Gould et al. (2021).

## 4. Computing the Projected-Error Gradient

This section describes how to compute the regret gradient $\nabla_{\hat{c}}\mathcal{R} = P_H e$ derived in Section 3 without explicitly forming the dense projection operator $P_H$. PEAR proceeds in three steps: (i) solve the forward problem to obtain primal and dual solutions, (ii) identify the active constraints and form the reduced Jacobian $J$, and (iii) compute $P_H e$ via a Schur-complement reduction over the active set.

### 4.1. Forward Pass: Active-Set Detection

We identify the active set from the primal–dual solution returned by the forward solver using a complementary slackness criterion. In our implementation, we solve (1) with OSQP (Stellato et al., 2020) after rewriting the constraints into $l \leq Gz \leq u$. Let $r^* = Gz^*$ denote the constraint residuals. We identify binding constraints via a complementary slackness test:

$$\begin{aligned} \mathcal{L} &= \{i \mid (r_i^* - l_i) < -y_i^*\}, \\ \mathcal{U} &= \{i \mid (u_i - r_i^*) < y_i^*\}, \end{aligned} \tag{16}$$

where $\mathcal{L}$ and $\mathcal{U}$ are the lower- and upper-active sets, and $\mathcal{A} = \mathcal{L} \cup \mathcal{U}$. Using $\mathcal{A}$, we construct the Jacobian $J$ as in (5).

### 4.2. Backward Pass: Efficient Gradient Computation

Our goal in the backward pass is to compute the regret gradient $g = \nabla_{\hat{c}}\mathcal{R}(\hat{c};c) = P_H e$ as derived in Theorem 3.5,

---

**Algorithm 1 PEAR**: Projected Error as Regret-gradient

---

**Input:** Predicted costs $\hat{c} = f_{\theta}(x)$, true costs $c$, problem data $(A, b, G, h)$
**Input:** Hessian $H = \nabla^2\phi(z^*)$ (or $H = \lambda I$ for LP), injection strength $\beta \in [0, 1]$
**Output:** Gradient signal $g = \nabla_{\hat{c}}\mathcal{R}(\hat{c};c)$

1: **// Forward: Active-Set Detection**
2: Solve (1) with cost $\hat{c}$ to obtain primal-dual pair $(z^*, y^*)$
3: Identify active set $\mathcal{A}$ via complementary slackness (16)
4: Form active constraint Jacobian $J \leftarrow \begin{bmatrix} A \\ G_{\mathcal{A}} \end{bmatrix}$
5: **// Backward: Compute $P_H e$ via Reduced System**
6: Compute prediction error $e \leftarrow \hat{c} - c$
7: Compute $x \leftarrow H^{-1}e$ and $r \leftarrow Jx$
8: Solve reduced system $(JH^{-1}J^{\top})v = r$ for $v$
9: Compute projected gradient $g \leftarrow x - H^{-1}J^{\top}v$
10: **// Optional: Normal injection for LP**
11: **if** $\beta > 0$ **then**
12:     Compute normal component $n \leftarrow H^{-1}J^{\top}v$
13:     $g \leftarrow g + \beta \cdot \frac{\|g\|}{\|n\|} \cdot n$
14: **end if**
15: **return** $g$

---

where $e = \hat{c} - c$. A direct computation of $P_H e$ would form the dense $n \times n$ projection operator $P_H$ explicitly, which becomes prohibitively expensive as dimension grows.

We instead compute $P_H e$ through a reduced linear system. The key insight is that under fixed active constraints, the primal perturbation $dz$ can be eliminated, leaving a reduced system determined only by the active constraints. This yields the same projected vector without forming $P_H$.

Under the regularity conditions and using the reduced KKT sensitivity system in (9), we set the cost perturbation to the prediction error $d\hat{c} = -e$ and obtain

$$\begin{bmatrix} H & J^{\top} \\ J & 0 \end{bmatrix} \begin{bmatrix} dz \\ dy \end{bmatrix} = \begin{bmatrix} e \\ 0 \end{bmatrix}, \tag{17}$$

where $H = \nabla^2\phi(z^*)$ and $J$ is the stacked Jacobian of the active constraints.

To eliminate $dz$, we use Schur complement reduction. From the first block of (17), we express

$$dz = H^{-1}(e - J^{\top}dy). \tag{18}$$

From the second block row of (17), we have $Jdz = 0$. Substituting (18) into this relation and defining $x = H^{-1}e$ and $r = Jx$, we obtain the reduced system

$$JH^{-1}J^{\top}dy = r. \tag{19}$$

This system has dimension $k \times k$ only and operates entirely within the reduced space of active multipliers, where $k =$

*Table 1.* **LP benchmarks.** Normalized regret (%, ↓) and training time (s, ↓), over 5 seeds. Best in **bold** and second best underlined.

| Task | Method | Deg 2 | | Deg 4 | | Deg 6 | | Deg 8 | |
|------|--------|-----------|----------|-----------|----------|-----------|----------|-----------|----------|
| | | Regret(%) | Time(s) | Regret(%) | Time(s) | Regret(%) | Time(s) | Regret(%) | Time(s) |
| Shortest Path | MSE | $0.135_{\pm 0.037}$ | $\mathbf{7.5_{\pm 0.8}}$ | $1.967_{\pm 0.565}$ | $\underline{7.5_{\pm 2.1}}$ | $6.561_{\pm 1.514}$ | $\mathbf{8.9_{\pm 1.5}}$ | $15.428_{\pm 3.085}$ | $\mathbf{6.1_{\pm 0.1}}$ |
| | SPO+ | $0.120_{\pm 0.046}$ | $48.7_{\pm 11.6}$ | $\mathbf{0.761_{\pm 0.242}}$ | $41.8_{\pm 10.8}$ | $\underline{2.281_{\pm 0.935}}$ | $44.1_{\pm 5.4}$ | $\underline{4.403_{\pm 1.491}}$ | $43.2_{\pm 7.5}$ |
| | PFYL | $0.177_{\pm 0.047}$ | $59.9_{\pm 3.2}$ | $0.946_{\pm 0.274}$ | $47.6_{\pm 12.0}$ | $2.435_{\pm 0.847}$ | $74.8_{\pm 22.9}$ | $5.236_{\pm 2.331}$ | $49.1_{\pm 12.1}$ |
| | DBB | $0.509_{\pm 0.109}$ | $116.4_{\pm 34.1}$ | $2.131_{\pm 0.597}$ | $139.3_{\pm 39.1}$ | $7.003_{\pm 1.936}$ | $98.7_{\pm 34.3}$ | $14.828_{\pm 3.086}$ | $85.2_{\pm 17.2}$ |
| | LAVA | $\mathbf{0.104_{\pm 0.036}}$ | $44.8_{\pm 8.1}$ | $0.851_{\pm 0.385}$ | $40.9_{\pm 3.9}$ | $2.483_{\pm 0.997}$ | $38.1_{\pm 2.4}$ | $6.189_{\pm 2.365}$ | $44.5_{\pm 6.0}$ |
| | PEAR (Ours) | $\mathbf{0.104_{\pm 0.038}}$ | $\underline{28.4_{\pm 5.4}}$ | $\underline{0.774_{\pm 0.268}}$ | $\mathbf{23.5_{\pm 0.4}}$ | $\mathbf{2.138_{\pm 0.762}}$ | $\underline{35.1_{\pm 4.9}}$ | $\mathbf{4.246_{\pm 1.106}}$ | $\underline{38.4_{\pm 2.5}}$ |
| Knapsack | MSE | $0.351_{\pm 0.020}$ | $\mathbf{422.8_{\pm 44.5}}$ | $0.922_{\pm 0.066}$ | $\mathbf{180.8_{\pm 6.2}}$ | $1.717_{\pm 0.226}$ | $\mathbf{111.7_{\pm 11.8}}$ | $2.285_{\pm 0.205}$ | $\mathbf{99.1_{\pm 15.8}}$ |
| | SPO+ | $\underline{0.291_{\pm 0.015}}$ | $600.0_{\pm 0.0}$ | $0.381_{\pm 0.036}$ | $600.0_{\pm 0.0}$ | $0.608_{\pm 0.066}$ | $600.0_{\pm 0.0}$ | $0.763_{\pm 0.075}$ | $600.0_{\pm 0.0}$ |
| | PFYL | $0.506_{\pm 0.025}$ | $600.0_{\pm 0.0}$ | $0.442_{\pm 0.042}$ | $600.0_{\pm 0.0}$ | $0.723_{\pm 0.049}$ | $600.0_{\pm 0.0}$ | $1.056_{\pm 0.147}$ | $600.0_{\pm 0.0}$ |
| | DBB | $1.852_{\pm 0.280}$ | $600.0_{\pm 0.0}$ | $0.910_{\pm 0.160}$ | $600.0_{\pm 0.0}$ | $1.040_{\pm 0.307}$ | $600.0_{\pm 0.0}$ | $1.081_{\pm 0.128}$ | $600.0_{\pm 0.0}$ |
| | LAVA | $0.438_{\pm 0.045}$ | $600.0_{\pm 0.0}$ | $0.478_{\pm 0.021}$ | $516.9_{\pm 81.1}$ | $0.820_{\pm 0.060}$ | $332.5_{\pm 111.0}$ | $1.351_{\pm 0.223}$ | $\underline{182.8_{\pm 42.5}}$ |
| | PEAR (Ours) | $\mathbf{0.287_{\pm 0.007}}$ | $\underline{567.9_{\pm 45.1}}$ | $\mathbf{0.315_{\pm 0.020}}$ | $\underline{305.2_{\pm 40.8}}$ | $\mathbf{0.388_{\pm 0.032}}$ | $\underline{261.7_{\pm 37.4}}$ | $\mathbf{0.437_{\pm 0.043}}$ | $271.7_{\pm 57.0}$ |

$p + |\mathcal{A}|$ consists of $p$ equalities and $|\mathcal{A}|$ active inequalities. Solving for $dy$, the projected gradient is then recovered as

$$g = P_H e = x - H^{-1} J^\top dy, \qquad (20)$$

with detailed derivation in Appendix B.4. In practice, we never form $H^{-1}$ explicitly and instead apply $H^{-1}$ through linear solves.

The computational advantage is substantial. Solving the $k \times k$ Schur complement system requires only matrix factorization and solving within the reduced space, whereas methods that differentiate through optimization solvers must handle full KKT systems. When the active set is sparse, this reduction yields significant efficiency gains without differentiating through solver iterations.

### 4.3. Linear Programs: Quadratic Smoothing and Normal Injection

In linear programming problems, the objective function lacks curvature and the solution map becomes piecewise constant. Over regions where the active set remains locally invariant, small cost perturbations do not alter the optimal solution. This creates plateaus where the regret gradient vanishes.

To obtain well-defined sensitivities while staying close to the original LP, we add a small quadratic regularization term $\phi(z) = \frac{\lambda}{2}\|z\|_2^2$ to our original problem (1), following Wilder et al. (2019). This yields $H = \lambda I \succ 0$, and the computation in Section 4.2 simplifies to

$$J J^\top v = J e, \qquad P_H e = \lambda^{-1}(e - J^\top v). \qquad (21)$$

However, even after smoothing, the pure regret gradient $g = P_H e$ can remain weak and nearly constant over large regions in LPs (Mandi et al., 2025). To stabilize learning in

*Table 2.* **LP benchmarks under noise.** Normalized regret at degree 8 under varying noise levels.

| Task | Method | Noise Level ($\epsilon$) | | |
|------|--------|-----------|-----------|-----------|
| | | 0.1 | 0.3 | 0.5 |
| Shortest Path | MSE | $15.59_{\pm 2.66}$ | $17.08_{\pm 2.47}$ | $20.69_{\pm 2.95}$ |
| | SPO+ | $\underline{4.74_{\pm 1.74}}$ | $6.79_{\pm 2.56}$ | $10.49_{\pm 2.47}$ |
| | PFYL | $5.44_{\pm 2.20}$ | $7.21_{\pm 2.81}$ | $\underline{10.44_{\pm 2.83}}$ |
| | DBB | $15.78_{\pm 3.85}$ | $17.95_{\pm 1.27}$ | $19.09_{\pm 1.89}$ |
| | LAVA | $6.60_{\pm 3.00}$ | $9.30_{\pm 3.78}$ | $12.46_{\pm 4.11}$ |
| | PEAR (Ours) | $\mathbf{4.50_{\pm 1.86}}$ | $\mathbf{5.88_{\pm 2.04}}$ | $\mathbf{10.38_{\pm 2.54}}$ |
| Knapsack | MSE | $2.33_{\pm 0.19}$ | $2.69_{\pm 0.19}$ | $3.24_{\pm 0.25}$ |
| | SPO+ | $\underline{0.81_{\pm 0.07}}$ | $\underline{1.12_{\pm 0.06}}$ | $\underline{1.77_{\pm 0.10}}$ |
| | PFYL | $1.09_{\pm 0.15}$ | $1.40_{\pm 0.12}$ | $2.00_{\pm 0.13}$ |
| | DBB | $1.23_{\pm 0.17}$ | $1.52_{\pm 0.26}$ | $2.07_{\pm 0.12}$ |
| | LAVA | $1.43_{\pm 0.21}$ | $1.85_{\pm 0.27}$ | $2.93_{\pm 0.38}$ |
| | PEAR (Ours) | $\mathbf{0.49_{\pm 0.05}}$ | $\mathbf{0.86_{\pm 0.05}}$ | $\mathbf{1.58_{\pm 0.08}}$ |

these regimes and obtain a more informative update signal, we use normal injection. PEAR provides the same active set quantities used to compute $g$, and the corresponding normal component can be computed from (21) as $n = \lambda^{-1} J^\top v$. We then form a controlled combination

$$g_{\text{inj}} = g + \beta \cdot \frac{\|g\|}{\|n\|} n, \qquad (22)$$

where $\beta \in [0, 1]$ controls the injection strength.

This update remains primarily driven by the regret signal, while selectively injecting normal space information when the tangent signal is weak. Scaling by $\|g\|$ preserves the relative magnitude of the injected term and prevents the normal correction from dominating the update. In flat regimes, this injection can enrich the gradient signal, helping the iterate reach neighborhoods where the active set changes and the regret signal becomes informative again.

*Table 3.* **Real-world QP.** Normalized regret (%, ↓), training time (s, ↓), and portfolio performance metrics over 5 seeds. Best in **bold** and second best underlined.

| Method | Regret(%) | Time(s) | Cum.Ret.(%) | Ann.Ret.(%) | Sharpe | MDD(%) |
|---|---|---|---|---|---|---|
| MSE | $90.80_{\pm2.05}$ | $\mathbf{33.0_{\pm9.5}}$ | $89.22_{\pm21.67}$ | $37.51_{\pm6.18}$ | $0.92_{\pm0.14}$ | $-40.00_{\pm4.49}$ |
| QPTH | $87.85_{\pm7.31}$ | $147.6_{\pm42.4}$ | $\underline{139.77_{\pm115.74}}$ | $\underline{47.88_{\pm26.99}}$ | $\underline{1.15_{\pm0.63}}$ | $-34.94_{\pm14.51}$ |
| CvxpyLayers | $\underline{87.36_{\pm7.36}}$ | $321.9_{\pm100.9}$ | $134.88_{\pm79.13}$ | $47.39_{\pm22.73}$ | $\underline{1.15_{\pm0.52}}$ | $-36.47_{\pm6.76}$ |
| PEAR (Ours) | $\mathbf{85.38_{\pm6.91}}$ | $\underline{122.3_{\pm38.6}}$ | $\mathbf{184.19_{\pm86.24}}$ | $\mathbf{58.84_{\pm19.49}}$ | $\mathbf{1.44_{\pm0.47}}$ | $\mathbf{-29.35_{\pm5.36}}$ |

## 5. Experiments

This section evaluates PEAR on two synthetic LP benchmarks and a real-world QP task, measuring decision quality via normalized regret and training efficiency. Beyond standard benchmarks, we test robustness under cost noise and constraint shifts at test time. All results are reported over 5 random seeds. Details in Appendix C and E.

### 5.1. Synthetic LP Benchmarks

**Setup.** Our experimental setup follows Berden et al. (2026) for two standard LP tasks—Shortest Path and Knapsack—while testing on additional degree configurations and different problem instances. *Shortest Path* predicts edge costs on a $5 \times 5$ grid (40 edges) and solves for the minimum-cost path. *Knapsack* predicts item values for a 100-item 0–1 knapsack instance and selects an optimal subset under a capacity constraint. For Knapsack, methods requiring continuous sensitivity (ours and LAVA) train on the LP relaxation; we report regret on the original problem. The cost vectors for both tasks are generated from feature vectors $x \sim \mathcal{N}(0, I_p)$ via a polynomial mapping (Elmachtoub & Grigas, 2022; Tang & Khalil, 2022):

$$c_{ij} = \left[ \frac{1}{3.5^{\text{deg}}} \left( \frac{1}{\sqrt{p}} (\mathcal{B}x_i)_j + 3 \right)^{\text{deg}} + 1 \right] \cdot \epsilon_{ij}, \quad (23)$$

where $\mathcal{B} \in \{0,1\}^{d \times p}$ is a random binary matrix and $p$ is the feature dimension. In the main experiments, we set $\epsilon_{ij} = 1$ (no noise). To evaluate robustness under model misspecification, we also test with multiplicative noise $\epsilon_{ij} \sim U(1 - \epsilon, 1 + \epsilon)$ for $\epsilon \in \{0.1, 0.3, 0.5\}$, which perturbs the cost-to-feature relationship and simulates scenarios where the true mapping is not perfectly captured by the polynomial model.

Baselines include MSE (two-stage) and representative DFL methods: SPO+ (Elmachtoub & Grigas, 2022), DBB (Pogančić et al., 2019), PFYL (Berthet et al., 2020), and LAVA (Berden et al., 2026). The first three are widely used baselines in the DFL literature, while LAVA is a recent solver-free approach that precomputes adjacent vertices of the LP polytope. All methods are implemented in PyEPO (Tang & Khalil, 2022) with recommended hyperparameter settings, using a linear predictor, Adam optimizer,

and early stopping when validation regret fails to improve by 1% for three consecutive evaluations. Training is capped at 600 seconds. Each task uses 1,000 training, 500 validation, and 500 test instances. We report normalized regret, defined as

$$\frac{\sum_{i=1}^{n_{\text{test}}} \left( c_i^\top z^*(\hat{c}_i) - c_i^\top z^*(c_i) \right)}{\sum_{i=1}^{n_{\text{test}}} |c_i^\top z^*(c_i)|}. \quad (24)$$

**Results.** Table 1 reports normalized regret and training time across polynomial degrees. On Shortest Path, our method matches or outperforms all baselines at every degree, with larger gains as nonlinearity increases. At degree 4, SPO+ achieves marginally lower regret, but PEAR trains roughly twice as fast. On Knapsack, our method achieves the lowest regret across all configurations, with the margin over the next-best method widening as degree increases. In terms of training time, our approach is consistently among the fastest DFL methods. On Knapsack, several baselines reach the timeout without converging, whereas our method terminates earlier in most cases. At degree 8, LAVA is slightly faster despite being solver-free, but its regret is more than three times higher than ours. Table 2 evaluates robustness under multiplicative noise in the cost generation process, simulating scenarios where the mapping from features to costs is not perfectly specified. Even under high noise levels ($\epsilon = 0.5$), PEAR consistently achieves the lowest regret on both tasks. Overall, these results demonstrate that computing regret gradients via tangent-space projection yields strong decision quality and training efficiency, with robustness to noise in the cost generation process.

### 5.2. Real-World QP: Mean–Variance Optimization

**Setup.** We build on the experimental framework of Hwang et al. (2025) for mean–variance optimization on S&P 100 constituents over 2010–2024. The predictor estimates 21-day ahead expected returns from a 63-day lookback window using DLinear (Zeng et al., 2023). The downstream optimizer constructs a long-only portfolio with risk aversion $\lambda = 2.0$. We rebalance every 21 trading days and include a 0.1% turnover cost.

We compare against the two-stage MSE baseline and two differentiable QP layers, QPTH (Amos & Kolter, 2017) and CvxpyLayers (Agrawal et al., 2019a). We evaluate normal-

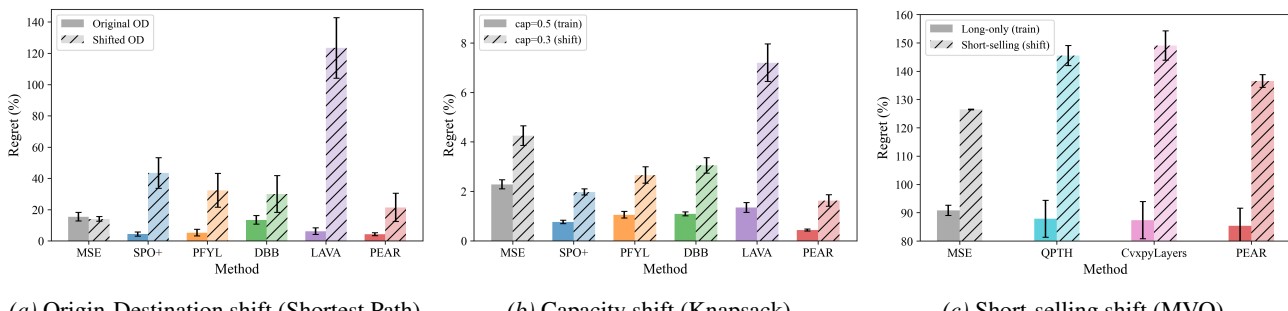

(a) Origin-Destination shift (Shortest Path)    (b) Capacity shift (Knapsack)    (c) Short-selling shift (MVO)

*Figure 2.* **Robustness under constraint shifts.** We train on an original train set and evaluate under shifted constraints. Our method maintains the lowest regret among DFL methods across all three tasks. Additional results are in Appendix F.

ized regret, cumulative and annualized returns, Sharpe ratio, and maximum drawdown.

**Batched computation.** In our implementation, the forward pass calls OSQP on each sample to obtain a primal–dual solution and identify the active constraints. We encode each active set as a binary mask indicating which inequality bounds are binding. The backward pass then proceeds entirely on GPU via batched linear algebra. Specifically, we (i) factorize $H = \lambda\Sigma$ via batched Cholesky decomposition, (ii) form the Schur complement using the binary masks, and (iii) solve for the dual variables and reconstruct $g = H^{-1}(e - J^\top v)$. This decouples the forward solver from the backward computation, allowing efficient gradient computation without a differentiable optimization layer.

**Results.** Table 3 summarizes performance on real-world mean–variance optimization. PEAR attains the lowest normalized regret and translates this gain into stronger realized portfolios, achieving the highest cumulative and annualized returns, the lowest maximum drawdown, and the best Sharpe ratio. PEAR is also faster than differentiable QP layers. While all approaches are KKT-based, differentiable optimization layers can become unstable near active-set boundaries; PEAR instead computes the regret-aligned projection directly, yielding more stable gradients that likely explain the gap in out-of-sample performance. See Appendix F.4 for additional analysis on model behavior during downturns.

### 5.3. Robustness Under Constraint Shifts

Most DFL evaluations assume a fixed feasible region at train and test time. In practice, however, constraints often change after deployment due to resource limits, policy updates, or risk controls. To evaluate robustness, we train models at degree 8 (the highest-degree setting) and test under modified constraints while keeping the cost-generation process fixed. Figure 2 presents three complementary shifts. Details on the modified problem structures are provided in Appendix D.

In *Shortest Path* (a), we swap the origin and destination nodes symmetrically across the grid. Existing DFL methods degrade dramatically under this shift, suggesting they memorize optimal paths rather than learning transferable cost structure. Our method exhibits the smallest degradation among DFL baselines, while MSE appears most robust because it treats all edges equally during training.

In *Knapsack* (b), we tighten the capacity constraint from 0.5 to 0.3. Our method again achieves the lowest regret among DFL methods. Unlike Shortest Path, MSE degrades more substantially here, likely because knapsack decisions depend on value rankings rather than absolute cost predictions, an inductive bias that DFL methods capture but MSE does not.

In *MVO* (c), we allow short-selling at test time while training on long-only constraints. Unlike the previous two tasks, this shift expands the feasible region rather than restricting it. Among DFL methods, PEAR achieves the lowest regret, while MSE again shows the strongest robustness. However, the performance gap between methods is smaller than in the combinatorial settings, likely because all KKT-based methods compute similar gradients for the QP objective.

Overall, PEAR degrades the least among DFL baselines under all three constraint shifts and consistently maintains the lowest regret. See Appendix H for additional discussion.

## 6. Conclusion

We introduced PEAR, a framework for Decision-Focused Learning that obtains regret gradients by projecting prediction error onto the tangent space of active constraints. This geometric view reveals that the DFL signal is the MSE gradient filtered of decision-irrelevant components. By computing gradients via a reduced linear system rather than differentiating through solvers, PEAR achieves computational efficiency and numerical stability. Experiments on LP benchmarks and portfolio optimization demonstrate that PEAR achieves superior decision quality with reduced training time and improved robustness under constraint shifts.

## Impact Statement

This work contributes to decision-focused learning by characterizing regret gradients geometrically and enabling their efficient computation without differentiating through optimization solvers. While such progress may have broad societal effects through downstream decision systems, we do not anticipate specific negative impacts beyond those common to predictive models used in optimization pipelines.

## Acknowledgments

This work was supported by the National Research Foundation of Korea (NRF) grant funded by the Korea government (MSIT) (No. RS-2025-24803208) and the Institute of Information & Communications Technology Planning & Evaluation (IITP) grant funded by the Korea government (MSIT) (No. RS-2020-II201336, Artificial Intelligence Graduate School Program (UNIST)).

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

# A. Additional Theory

## A.1. Regularity and Local Differentiability

This appendix states standard conditions under which the solution mapping $\hat{c} \mapsto z^*(\hat{c})$ is locally single-valued and differentiable and the active set remains locally stable. Under Assumptions 3.1–3.3, classical sensitivity results imply that there exists a neighborhood of $\hat{c}$ on which the active set $\mathcal{A}(\hat{c})$ is constant and the KKT system admits a unique primal–dual solution; see, e.g., Fiacco (1983); Bonnans & Shapiro (2013). Consequently, on such a neighborhood the inequality-constrained problem locally behaves as the reduced equality-constrained problem in Section 3.3, and the implicit-function-based sensitivity analysis is valid.

## A.2. Derivation of the Reduced KKT Sensitivity

For completeness, we derive Equations (10) and (11) from Equation (9) by block elimination. Let $S = JH^{-1}J^\top$. Since $H \succ 0$ and $J$ has full row rank, $S$ is invertible. From the first block row of Equation (9),

$$Hdz + J^\top dy = -d\hat{c} \quad \Rightarrow \quad dz = -H^{-1}d\hat{c} - H^{-1}J^\top dy.$$

Substituting into the second block row $Jdz = 0$ gives

$$-JH^{-1}d\hat{c} - JH^{-1}J^\top dy = 0 \quad \Rightarrow \quad dy = -S^{-1}JH^{-1}d\hat{c}.$$

Plugging back,

$$dz = -H^{-1}d\hat{c} + H^{-1}J^\top S^{-1}JH^{-1}d\hat{c} = -P_H\, d\hat{c},$$

with $P_H$ as in Equation (11).

# B. Proofs

## B.1. Proof of Proposition 3.4

*Proof.* Let $S = JH^{-1}J^\top$, which is invertible under Assumption 3.1 and Assumption 3.2.

*(i)* We show $JP_H = 0$. By definition of $P_H$,

$$\begin{aligned}
JP_H &= JH^{-1} - JH^{-1}J^\top S^{-1}JH^{-1} \\
&= JH^{-1} - SS^{-1}JH^{-1} \\
&= JH^{-1} - JH^{-1} = 0.
\end{aligned} \tag{25}$$

For $P_H J^\top = 0$, note that $P_H$ is symmetric:

$$P_H^\top = H^{-1} - H^{-1}J^\top(JH^{-1}J^\top)^{-1}JH^{-1} = P_H, \tag{26}$$

since $H^{-1}$ and $S^{-1}$ are both symmetric. Thus $P_H J^\top = (JP_H)^\top = 0$.

*(ii)* For idempotency of $\Pi_H$:

$$\begin{aligned}
\Pi_H^2 &= (I - H^{-1}J^\top S^{-1}J)^2 \\
&= I - 2H^{-1}J^\top S^{-1}J + H^{-1}J^\top S^{-1}JH^{-1}J^\top S^{-1}J \\
&= I - 2H^{-1}J^\top S^{-1}J + H^{-1}J^\top S^{-1}SS^{-1}J \\
&= I - 2H^{-1}J^\top S^{-1}J + H^{-1}J^\top S^{-1}J \\
&= I - H^{-1}J^\top S^{-1}J = \Pi_H.
\end{aligned} \tag{27}$$

For $H$-self-adjointness, we show $\langle u, \Pi_H v\rangle_H = \langle \Pi_H u, v\rangle_H$ where $\langle a, b\rangle_H := a^\top Hb$:

$$\begin{aligned}
\langle u, \Pi_H v\rangle_H &= u^\top H\Pi_H v \\
&= u^\top H(I - H^{-1}J^\top S^{-1}J)v \\
&= u^\top(H - J^\top S^{-1}J)v.
\end{aligned} \tag{28}$$

Since $H$ and $S^{-1}$ are symmetric, the matrix $H - J^\top S^{-1} J$ is symmetric, hence $\langle u, \Pi_H v \rangle_H = \langle \Pi_H u, v \rangle_H$.

*(iii)* Direct expansion:

$$\begin{aligned}
\Pi_H H^{-1} &= (I - H^{-1} J^\top S^{-1} J) H^{-1} \\
&= H^{-1} - H^{-1} J^\top S^{-1} J H^{-1} = P_H.
\end{aligned} \tag{29}$$

$\square$

## B.2. Proof of Theorem 3.5

*Proof.* Recall the regret function $\mathcal{R}(\hat{c}; c) = f(z^*(\hat{c}); c) - f(z^*(c); c)$, where $f(z; c) = \phi(z) + c^\top z$. Since $z^*(c)$ does not depend on $\hat{c}$, we have

$$\nabla_{\hat{c}} \mathcal{R}(\hat{c}; c) = \nabla_{\hat{c}} f(z^*(\hat{c}); c).$$

*Step 1: Chain rule.* On any neighborhood where the active set is fixed, the solution map is differentiable (Appendix A.1). As shown in Eq. (4), applying the chain rule yields

$$\begin{aligned}
\nabla_{\hat{c}} \mathcal{R}(\hat{c}; c) &= \left( \frac{\partial z^*}{\partial \hat{c}} \right)^\top \nabla_z f(z; c) \Big|_{z = z^*(\hat{c})} \\
&= \left( \frac{\partial z^*}{\partial \hat{c}} \right)^\top (\nabla \phi(z^*) + c).
\end{aligned} \tag{30}$$

*Step 2: Substitute solution sensitivity.* From Equation (10), $\frac{\partial z^*}{\partial \hat{c}} = -P_H$. Since $P_H$ is symmetric (Appendix B.1), we obtain

$$\nabla_{\hat{c}} \mathcal{R}(\hat{c}; c) = -P_H (\nabla \phi(z^*) + c).$$

*Step 3: Apply KKT stationarity.* From the reduced KKT stationarity Equation (7),

$$\nabla \phi(z^*) + \hat{c} + J^\top y^* = 0 \quad \Rightarrow \quad \nabla \phi(z^*) + c = -(\hat{c} - c) - J^\top y^*.$$

Substituting gives

$$\nabla_{\hat{c}} \mathcal{R}(\hat{c}; c) = P_H (\hat{c} - c) + P_H J^\top y^*.$$

*Step 4: Remove normal component.* By Proposition 3.4, $P_H J^\top = 0$, so the second term vanishes and

$$\nabla_{\hat{c}} \mathcal{R}(\hat{c}; c) = P_H (\hat{c} - c) = P_H e.$$

$\square$

## B.3. Proof of Corollary 3.6

*Proof.* We prove the two statements separately.

**Proof of Equation (14).** Let $\delta_N \in \mathcal{N} = \text{range}(J^\top)$. Then there exists a vector $v$ such that $\delta_N = J^\top v$. By the projection properties in Proposition 3.4, we have

$$P_H J^\top = 0,$$

and therefore

$$P_H \delta_N = P_H J^\top v = 0.$$

**Proof of Equation (15).** Assume that the active set remains unchanged when replacing $\hat{c}$ by $\hat{c} + \alpha \delta_N$ for any scalar $\alpha$. Under this assumption, the same Jacobian $J$ and projection operator $P_H$ apply. By Theorem 3.5, the regret gradient is given by

$$\nabla_{\hat{c}} \mathcal{R}(\hat{c}; c) = P_H (\hat{c} - c).$$

Hence,

$$\nabla_{\hat{c}}\mathcal{R}(\hat{c}+\alpha\delta_N;c) = P_H\big((\hat{c}+\alpha\delta_N)-c\big) = P_H(\hat{c}-c)+\alpha P_H\delta_N.$$

Using Equation (14), the second term vanishes, which yields

$$\nabla_{\hat{c}}\mathcal{R}(\hat{c}+\alpha\delta_N;c) = P_H(\hat{c}-c).$$

This proves the invariance of the regret gradient under normal perturbations. $\square$

### B.4. Proof of Section 4.2

*Proof.* We derive the reduced system by explicitly solving the linearized KKT system defined in Equation (17):

$$\begin{bmatrix} H & J^\top \\ J & 0 \end{bmatrix}\begin{bmatrix} dz \\ dy \end{bmatrix} = \begin{bmatrix} e \\ 0 \end{bmatrix}. \tag{31}$$

Writing out the block equations gives:

$$Hdz + J^\top dy = e, \tag{32}$$
$$Jdz = 0. \tag{33}$$

Since $H$ is positive definite (Assumption 3.1), it is invertible. Solving (32) for the primal perturbation $dz$ yields:

$$dz = H^{-1}(e - J^\top dy) = H^{-1}e - H^{-1}J^\top dy. \tag{34}$$

To eliminate $dz$, we substitute (34) into the constraint equation (33):

$$J\left(H^{-1}e - H^{-1}J^\top dy\right) = 0. \tag{35}$$

Rearranging terms to isolate the dual variable $dy$ leads to the Schur complement system:

$$(JH^{-1}J^\top)dy = JH^{-1}e. \tag{36}$$

Letting $v = dy$ denote the dual perturbation, and defining the intermediate variables $x = H^{-1}e$ and $r = Jx$, this system becomes

$$(JH^{-1}J^\top)v = r, \tag{37}$$

which matches the reduced Schur complement system in Equation (19).

Finally, substituting $v$ back into (34) yields $dz = H^{-1}e - H^{-1}J^\top v$. Since the solution of the linearized KKT system satisfies $dz = P_He$, we identify the projected gradient as $g = P_He = dz$, and hence

$$g = x - H^{-1}J^\top v, \tag{38}$$

which is Equation (20). Thus, solving the reduced Schur complement system suffices to compute $P_He$ without explicitly forming $P_H$. $\square$

## C. Optimization Problem Details

This appendix provides full specifications and data generation procedures for the benchmark problems and real-world portfolio optimization problem used in our experiments.

### C.1. Shortest Path

We consider the standard shortest-path benchmark used in prior work (Elmachtoub & Grigas, 2022; Tang & Khalil, 2022). The problem is defined on a directed $5 \times 5$ grid graph $G = (V, E)$, where each node $(i, j) \in V$ is connected to its right and downward neighbors. The edge set is

$$E = \big\{\,((i,j),(i+1,j)) \mid 0 \le i < 4,\ 0 \le j \le 4\,\big\} \cup \big\{\,((i,j),(i,j+1)) \mid 0 \le i \le 4,\ 0 \le j < 4\,\big\},$$

yielding $|E| = 40$ edges and $|V| = 25$ nodes, with source $s = (0,0)$ (top-left) and target $t = (4,4)$ (bottom-right). Let $c_{uv}$ be the cost of edge $(u,v)$ and $w_{uv}$ the decision variable indicating whether edge $(u,v)$ lies on the path, and let $b \in \mathbb{R}^{|V|}$ be the demand vector with $b_s = -1$, $b_t = 1$, and $b_v = 0$ for every other node. The shortest path problem is the integer program whose linear relaxation is

$$
\begin{aligned}
\min_{w} \quad & \sum_{(u,v) \in E} c_{uv}\, w_{uv} \\
\text{subject to} \quad & \sum_{u\,:\,(u,v) \in E} w_{uv} - \sum_{u\,:\,(v,u) \in E} w_{vu} = b_v, \quad \forall v \in V \\
& 0 \le w_{uv} \le 1, \quad \forall (u,v) \in E
\end{aligned}
\tag{39}
$$

The first sum runs over edges entering $v$ and the second over edges leaving $v$, so the flow-balance constraints route one unit of flow from the source $s$ to the target $t$.

## C.2. Knapsack

We consider a binary knapsack problem with $n = 100$ items. Let $z_i = 1$ if item $i$ is selected, and 0 otherwise. The problem is formulated as:

$$
\begin{aligned}
\max_{z} \quad & \sum_{i=1}^{n} c_i z_i \\
\text{subject to} \quad & \sum_{i=1}^{n} w_i z_i \le C \\
& z_i \in \{0,1\}, \quad \forall i \in \{1, \ldots, n\}
\end{aligned}
\tag{40}
$$

The objective maximizes the total value of selected items. The constraint ensures that the total weight does not exceed capacity $C$.

**LP Relaxation.** Since the original problem is an integer program, methods requiring continuous relaxation (PEAR and LAVA) solve the LP relaxation during training:

$$
\begin{aligned}
\max_{z} \quad & c^\top z \\
\text{subject to} \quad & w^\top z \le C \\
& 0 \le z_i \le 1, \quad \forall i \in \{1, \ldots, n\}
\end{aligned}
\tag{41}
$$

All methods are evaluated on the original integer problem.

## C.3. Mean-Variance Portfolio Optimization

We study portfolio optimization using historical daily returns from S&P 100 constituents over 2010–2024.

**Problem Formulation.** Let $w \in \mathbb{R}^n$ represent portfolio weights. The mean-variance optimization problem is:

$$
\begin{aligned}
\min_{w} \quad & \frac{\lambda}{2}\, w^\top \Sigma w - \mu^\top w \\
\text{subject to} \quad & \mathbf{1}^\top w = 1, \quad w \ge 0
\end{aligned}
\tag{42}
$$

where $\mu \in \mathbb{R}^n$ is the expected return vector, $\Sigma \in \mathbb{R}^{n \times n}$ is the covariance matrix, and $\lambda = 2.0$ is the risk aversion parameter. The constraints enforce a fully-invested long-only portfolio.

**Data Construction.** We use daily adjusted close prices to compute returns. The prediction model takes a 63-day lookback window of historical returns as input and forecasts 21-day ahead returns. Expected returns $\mu$ are computed as the mean of predicted daily returns. The covariance matrix $\Sigma$ is estimated from the combined 84-day window (63-day history + 21-day prediction) with spectral shifting to ensure positive definiteness.

# D. Optimization Problems for Constraint Shift Experiments

This appendix details the modified optimization problems used in the constraint shift experiments. These experiments evaluate how well DFL methods generalize when the feasible region changes between training and test time, while the cost structure remains the same.

### D.1. Shortest Path: Direction Generalization

We additionally evaluate generalization across routing directions. Each configuration shares the $5 \times 5$ grid node set and the cost-generation procedure of Appendix C.1, but uses its own directed edge set and origin–destination pair. The optimization has the same form as the linear program in Appendix C.1, solved on the configuration-specific directed graph; the base problem is the forward configuration below.

**Experimental Setup.**

- **Forward (Training):** source $(0,0)$, target $(4,4)$, with edges directed rightward and downward.

- **Cross (Test):** source $(4,0)$, target $(0,4)$, with edges directed rightward and upward.

Only the directed edge set and the origin–destination pair differ between the two configurations; the grid nodes and the cost-generation procedure are identical. This tests whether learned edge-cost predictions transfer to a different routing direction on the same grid.

### D.2. Knapsack: Capacity Generalization

We modify the knapsack problem from Appendix C.2 by varying the capacity constraint. The parameterized problem is:

$$
\begin{aligned}
\max_{z} \quad & \sum_{i=1}^{n} c_i z_i \\
\text{subject to} \quad & \sum_{i=1}^{n} w_i z_i \leq C(\rho) \\
& z_i \in \{0,1\}, \quad \forall i \in \{1,\ldots,n\}
\end{aligned}
\tag{43}
$$

where the capacity is defined as a fraction of total weight:

$$
C(\rho) = \rho \cdot \sum_{i=1}^{n} w_i.
\tag{44}
$$

**Experimental Setup.** Models are trained with $\rho_{\text{train}} = 0.5$ and evaluated on capacities $\rho_{\text{test}} \in \{0.3, 0.5, 0.7, 0.9\}$. Lower capacity ratios create tighter constraints (fewer items can fit), while higher ratios relax the constraint (more items can fit). This tests whether methods learn cost predictions that generalize across different constraint tightness levels.

### D.3. Portfolio Optimization: Short Selling Generalization

We modify the mean-variance optimization problem from Appendix C.3 by relaxing the long-only constraint to allow short selling. The parameterized problem is:

$$
\begin{aligned}
\min_{z} \quad & \frac{\lambda}{2} z^{\top} \Sigma z - \mu^{\top} z \\
\text{subject to} \quad & \mathbf{1}^{\top} z = 1 \\
& z \geq \ell
\end{aligned}
\tag{45}
$$

where $\ell \in \mathbb{R}$ is the lower bound on individual asset weights.

**Experimental Setup.** Models are trained with $\ell_{\text{train}} = 0$ (long-only portfolio) and evaluated on $\ell_{\text{test}} \in \{-0.1, -0.3, -0.5, -1.0\}$. Negative lower bounds allow short positions up to the specified limit. For example, $\ell = -0.1$ permits shorting up to 10% of portfolio value in any single asset. This tests whether return predictions learned under long-only constraints remain useful when the optimization gains additional flexibility through short selling.

## D.4. Summary of Constraint Shifts

Table 4 summarizes the constraint modifications across all three problem domains.

*Table 4.* Summary of constraint shift experiments. All experiments use the same cost/return prediction model trained under the training constraint, evaluated across multiple test constraints.

| Problem | Modified Constraint | Training | Test |
|---|---|---|---|
| Knapsack | Capacity ratio $\rho$ | 0.5 | $\{0.3, 0.5, 0.7, 0.9\}$ |
| Shortest Path | Origin-destination pair $(v_s, v_t)$ | $(0, 0) \to (4, 4)$ | $(0, 4) \to (4, 0)$ |
| Portfolio | Weight lower bound $\ell$ | 0 | $\{-0.1, -0.3, -0.5, -1.0\}$ |

# E. Experimental Details

Our code is available at `https://github.com/FinJun/PEAR`. All experiments are conducted on a single NVIDIA RTX A5000 GPU with an Intel Xeon Silver 4210R CPU, and repeated over 5 random seeds $\{0, 1, 2, 3, 4\}$ with mean and standard deviation reported.

## E.1. LP benchmarks Experiments (Shortest Path, Knapsack)

Following Elmachtoub & Grigas (2022), we generate 1,000 training, 500 validation, and 500 test samples, varying the polynomial degree $d \in \{2, 4, 6, 8\}$ to control problem difficulty. We use a single-layer linear network mapping 5-dimensional features to edge costs (40 dimensions for Shortest Path) or item values (100 dimensions for Knapsack). The parameter $\beta$ for normal injection in PEAR was fixed at 0.1 throughout all experiments.

## E.2. Real-world QP Experiments (Portfolio Optimization)

We use a chronological 70:10:20 split for train, validation, and test sets. The prediction model is DLinear (Zeng et al., 2023) with RevIN (Kim et al., 2021) normalization, using a lookback window of $L = 63$ days and prediction horizon of $H = 21$ days. All methods are trained with the Adam optimizer using learning rate $10^{-3}$ and batch size 64, with learning rate reduction (factor 0.5, patience 3) and early stopping (patience 10) based on validation regret.

For evaluation, portfolios are rebalanced every 21 trading days with a transaction cost of 0.1% applied to turnover. We report cumulative return $\prod_{t=1}^{T}(1 + r_t) - 1$, annualized return and volatility (scaled by $\sqrt{252}$), Sharpe ratio assuming zero risk-free rate, maximum drawdown, and average turnover at each rebalancing.

# F. Additional Results

We evaluate generalization under constraint shifts, where test-time constraints differ from training. This setting is practically important since real-world deployment often involves changing requirements.

## F.1. Capacity Shifts (Knapsack)

We train models with capacity ratio 0.5 and evaluate on ratios $\{0.3, 0.5, 0.7, 0.9\}$. As shown in Table 5, PEAR consistently achieves the best performance across all polynomial degrees and capacity ratios. Surrogate-based methods (SPO+, PFYL) degrade significantly at higher capacities (0.9), while PEAR maintains robust performance. DBB exhibits particularly poor generalization at low polynomial degrees.

## F.2. Direction Shifts (Shortest Path)

We train models on one source-target pair and evaluate on a different direction. Table 6 reveals an interesting pattern: MSE achieves the best performance across all degrees, with PEAR as the second best. This suggests that task-specific DFL objectives can overfit to training constraint structures, whereas prediction-focused training (MSE) and geometry-aware methods (PEAR) generalize better to unseen constraint configurations.

*Table 5.* **Capacity generalization (Knapsack).** Normalized regret (%, ↓) under varying capacity ratios. Best in **bold**, second best underlined. *Training capacity.

| Method | Degree 2 | | | | Degree 4 | | | |
|---|---|---|---|---|---|---|---|---|
| | 0.3 | 0.5* | 0.7 | 0.9 | 0.3 | 0.5* | 0.7 | 0.9 |
| MSE | $0.449_{\pm0.011}$ | $0.350_{\pm0.017}$ | $\underline{0.255}_{\pm0.014}$ | $\underline{0.141}_{\pm0.014}$ | $1.269_{\pm0.082}$ | $0.922_{\pm0.059}$ | $1.037_{\pm0.123}$ | $\underline{1.557}_{\pm0.219}$ |
| SPO+ | $\mathbf{0.373}_{\pm\mathbf{0.006}}$ | $\underline{0.291}_{\pm0.013}$ | $0.260_{\pm0.012}$ | $0.444_{\pm0.056}$ | $\mathbf{0.542}_{\pm\mathbf{0.034}}$ | $0.381_{\pm0.032}$ | $0.834_{\pm0.087}$ | $2.520_{\pm0.216}$ |
| PFYL | $0.610_{\pm0.024}$ | $0.506_{\pm0.022}$ | $0.937_{\pm0.103}$ | $5.109_{\pm0.428}$ | $0.628_{\pm0.060}$ | $0.441_{\pm0.038}$ | $0.922_{\pm0.145}$ | $3.792_{\pm0.620}$ |
| DBB | $2.514_{\pm0.264}$ | $1.852_{\pm0.250}$ | $2.093_{\pm0.217}$ | $6.155_{\pm0.629}$ | $1.003_{\pm0.118}$ | $0.910_{\pm0.143}$ | $1.907_{\pm0.213}$ | $5.542_{\pm0.342}$ |
| LAVA | $0.636_{\pm0.051}$ | $0.439_{\pm0.039}$ | $0.537_{\pm0.021}$ | $2.191_{\pm0.186}$ | $1.346_{\pm0.074}$ | $0.477_{\pm0.018}$ | $\underline{0.644}_{\pm0.053}$ | $3.337_{\pm0.770}$ |
| PEAR | $\underline{0.420}_{\pm0.041}$ | $\mathbf{0.287}_{\pm\mathbf{0.006}}$ | $\mathbf{0.222}_{\pm\mathbf{0.010}}$ | $\mathbf{0.140}_{\pm\mathbf{0.029}}$ | $\underline{0.589}_{\pm0.057}$ | $\mathbf{0.315}_{\pm\mathbf{0.017}}$ | $\mathbf{0.355}_{\pm\mathbf{0.029}}$ | $\mathbf{0.590}_{\pm\mathbf{0.254}}$ |

| Method | Degree 6 | | | | Degree 8 | | | |
|---|---|---|---|---|---|---|---|---|
| | 0.3 | 0.5* | 0.7 | 0.9 | 0.3 | 0.5* | 0.7 | 0.9 |
| MSE | $2.629_{\pm0.303}$ | $1.717_{\pm0.202}$ | $1.946_{\pm0.164}$ | $2.909_{\pm0.171}$ | $4.252_{\pm0.394}$ | $2.285_{\pm0.183}$ | $2.036_{\pm0.182}$ | $2.732_{\pm0.204}$ |
| SPO+ | $1.187_{\pm0.066}$ | $0.608_{\pm0.059}$ | $1.225_{\pm0.109}$ | $3.416_{\pm0.233}$ | $1.975_{\pm0.123}$ | $0.763_{\pm0.067}$ | $1.194_{\pm0.117}$ | $3.067_{\pm0.243}$ |
| PFYL | $1.528_{\pm0.082}$ | $0.723_{\pm0.044}$ | $1.165_{\pm0.254}$ | $3.783_{\pm1.056}$ | $2.658_{\pm0.331}$ | $1.056_{\pm0.132}$ | $1.255_{\pm0.113}$ | $3.308_{\pm0.351}$ |
| DBB | $1.437_{\pm0.298}$ | $1.040_{\pm0.275}$ | $2.026_{\pm0.330}$ | $4.864_{\pm0.472}$ | $2.156_{\pm0.220}$ | $1.081_{\pm0.114}$ | $1.849_{\pm0.117}$ | $4.256_{\pm0.218}$ |
| LAVA | $3.742_{\pm0.259}$ | $0.820_{\pm0.053}$ | $0.996_{\pm0.228}$ | $4.721_{\pm1.124}$ | $7.196_{\pm0.758}$ | $1.351_{\pm0.200}$ | $1.541_{\pm0.498}$ | $5.397_{\pm1.086}$ |
| PEAR | $\mathbf{0.986}_{\pm\mathbf{0.088}}$ | $\mathbf{0.388}_{\pm\mathbf{0.029}}$ | $\mathbf{0.478}_{\pm\mathbf{0.036}}$ | $\mathbf{1.043}_{\pm\mathbf{0.224}}$ | $\mathbf{1.628}_{\pm\mathbf{0.233}}$ | $\mathbf{0.437}_{\pm\mathbf{0.038}}$ | $\mathbf{0.554}_{\pm\mathbf{0.058}}$ | $\mathbf{1.232}_{\pm\mathbf{0.197}}$ |

*Table 6.* **Direction generalization (Shortest Path).** Normalized regret (%, ↓) under constraint shift. Best in **bold**, second best underlined.

| Method | Degree 2 | Degree 4 | Degree 6 | Degree 8 |
|---|---|---|---|---|
| MSE | $\mathbf{0.14}_{\pm\mathbf{0.03}}$ | $\mathbf{1.95}_{\pm\mathbf{0.44}}$ | $\mathbf{6.53}_{\pm\mathbf{1.30}}$ | $\mathbf{14.00}_{\pm\mathbf{1.47}}$ |
| SPO+ | $9.38_{\pm4.40}$ | $22.14_{\pm8.43}$ | $29.03_{\pm6.71}$ | $43.40_{\pm9.83}$ |
| PFYL | $9.38_{\pm2.37}$ | $21.05_{\pm6.26}$ | $23.35_{\pm6.15}$ | $32.33_{\pm10.89}$ |
| DBB | $3.59_{\pm0.72}$ | $8.95_{\pm6.08}$ | $22.55_{\pm8.69}$ | $31.06_{\pm5.78}$ |
| LAVA | $16.84_{\pm3.12}$ | $45.01_{\pm8.40}$ | $78.73_{\pm13.33}$ | $123.40_{\pm19.35}$ |
| PEAR | $\underline{0.32}_{\pm0.37}$ | $\underline{3.63}_{\pm2.08}$ | $\underline{7.07}_{\pm1.90}$ | $\underline{21.42}_{\pm9.02}$ |

## F.3. Lower Bound Shifts (Portfolio Optimization)

We train models with long-only constraints (lb = 0) and evaluate under relaxed lower bounds allowing short-selling. Table 7 shows that MSE achieves the best generalization, with PEAR as the closest DFL method. Differentiable optimization methods (QPTH, CvxpyLayers) show degraded performance as constraints deviate further from training conditions.

*Table 7.* **Lower bound generalization (MVO).** Normalized regret (%, ↓) under varying lower bound constraints. Best in **bold**, second best underlined.

| Method | lb=-0.1 | lb=-0.3 | lb=-0.5 | lb=-1.0 |
|---|---|---|---|---|
| MSE | $\mathbf{119.9}_{\pm\mathbf{0.2}}$ | $\mathbf{126.4}_{\pm\mathbf{0.1}}$ | $\mathbf{130.6}_{\pm\mathbf{0.1}}$ | $\mathbf{138.3}_{\pm\mathbf{0.1}}$ |
| QPTH | $129.3_{\pm2.4}$ | $145.5_{\pm4.0}$ | $152.4_{\pm5.7}$ | $162.4_{\pm6.6}$ |
| CvxpyLayers | $129.8_{\pm2.5}$ | $149.0_{\pm5.8}$ | $156.9_{\pm6.8}$ | $167.6_{\pm7.1}$ |
| PEAR | $\underline{124.1}_{\pm3.4}$ | $\underline{136.5}_{\pm2.5}$ | $\underline{142.6}_{\pm1.7}$ | $\underline{153.4}_{\pm0.9}$ |

## F.4. Portfolio Performance Analysis

Beyond regret minimization (Table 3), we evaluate whether PEAR's gradient estimation translates to superior real-world portfolio performance. In mean-variance optimization, balancing returns against risk is crucial—high returns alone are insufficient if the portfolio exhibits excessive volatility during market turbulence. Figure 3 shows cumulative portfolio value over the out-of-sample period, where PEAR achieves the highest terminal value, demonstrating that improved training regret translates to superior realized returns. More importantly, PEAR's advantage extends beyond raw returns. Figure 4 examines portfolio behavior during the two most severe drawdown periods (days 80–180 and 560–640). In both cases, PEAR exhibits the most stable trajectory, demonstrating superior robustness to market downturns compared to differentiable optimization methods. This suggests that PEAR learns predictions that better capture the risk-return trade-off encoded in the

mean-variance objective.

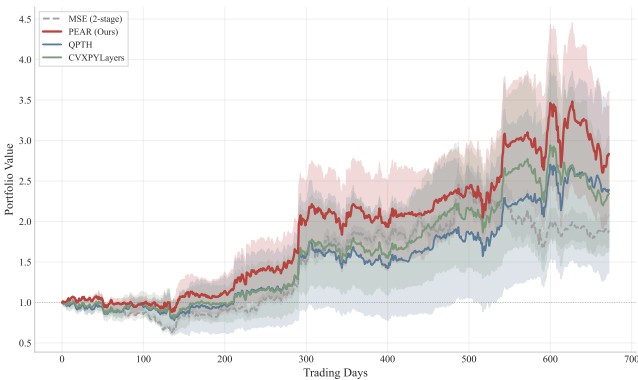

*Figure 3.* **Portfolio cumulative returns.** Portfolio value over the test period. PEAR achieves the highest value among all methods.

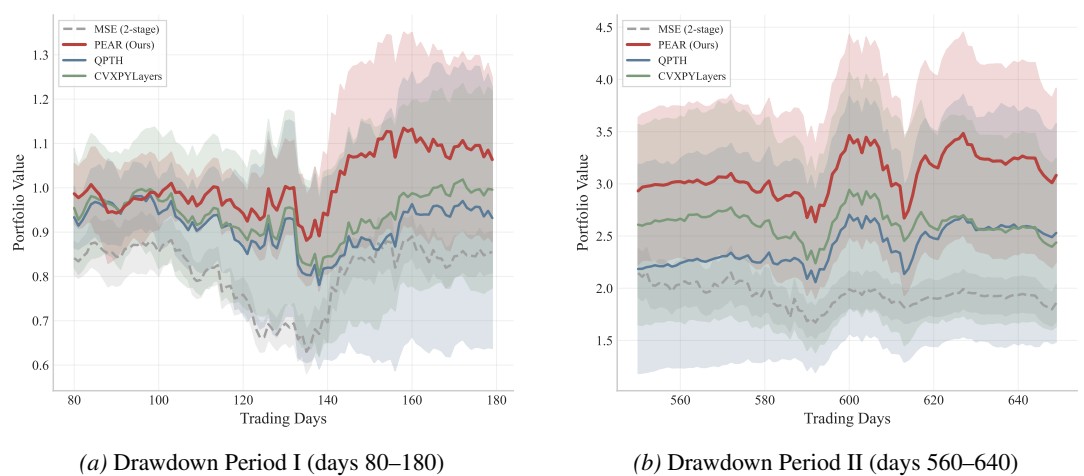

*(a)* Drawdown Period I (days 80–180)          *(b)* Drawdown Period II (days 560–640)

*Figure 4.* **Drawdown analysis.** Portfolio value during the two most severe market downturns. PEAR demonstrates the most stable behavior, indicating robust risk management.

## G. Additional Experiments

This appendix presents additional empirical results that complement those in the main paper. Appendix G.1 analyzes sensitivity to PEAR's two LP-specific hyperparameters, and Appendix G.2 evaluates PEAR on a real-world knapsack.

### G.1. Hyperparameter Ablations on LP Benchmarks

For LP problems, PEAR introduces two hyperparameters, the quadratic smoothing strength $\lambda$ and the normal injection coefficient $\beta$. We isolate the sensitivity of PEAR to each on the two LP benchmarks. All other settings follow Appendix E.1, and we report normalized regret over 5 seeds. The default used in the main experiments is $(\lambda, \beta) = (0.1, 0.1)$.

**Normal Injection.** We sweep $\beta \in \{0, 0.05, 0.1, 0.2, 0.5\}$ with $\lambda = 0.1$ fixed. Table 8 reports the normalized regret. On Shortest Path the choice of $\beta$ has only a mild effect. $\beta = 0$ is already competitive and attains the best regret at the hardest degree (Deg 8), while small positive $\beta$ is marginally better at the intermediate degrees. On Knapsack the effect is more pronounced—a small amount of normal information consistently helps, with $\beta = 0.05$ best at the higher degrees (Deg 6 and 8) and $\beta = 0.1$ best at Deg 4, whereas the pure gradient is the weakest of the small-$\beta$ settings. Large $\beta$ (0.5) degrades performance on Knapsack, worsening monotonically with the polynomial degree. PEAR is otherwise stable for $\beta \in [0.05, 0.2]$, and we use $\beta = 0.1$ as the default.

*Table 8.* **Normal-injection ablation.** Normalized regret (%, ↓) under varying $\beta$, with $\lambda = 0.1$ fixed. Best $\beta$ per degree in **bold**, second best underlined.

| Task | $\beta$ | Deg 2 | Deg 4 | Deg 6 | Deg 8 |
|---|---|---|---|---|---|
| Shortest Path | 0 | $0.111_{\pm 0.030}$ | $0.777_{\pm 0.216}$ | $2.314_{\pm 0.997}$ | $\mathbf{4.215_{\pm 1.294}}$ |
| | 0.05 | $0.119_{\pm 0.014}$ | $\mathbf{0.751_{\pm 0.226}}$ | $2.257_{\pm 1.019}$ | $5.763_{\pm 3.971}$ |
| | 0.1 | $\underline{0.104_{\pm 0.034}}$ | $0.774_{\pm 0.239}$ | $\mathbf{2.138_{\pm 0.681}}$ | $\underline{4.246_{\pm 0.989}}$ |
| | 0.2 | $0.112_{\pm 0.035}$ | $0.801_{\pm 0.249}$ | $\underline{2.162_{\pm 0.726}}$ | $4.574_{\pm 1.915}$ |
| | 0.5 | $\mathbf{0.096_{\pm 0.033}}$ | $0.811_{\pm 0.239}$ | $2.277_{\pm 0.963}$ | $4.329_{\pm 1.633}$ |
| Knapsack | 0 | $0.299_{\pm 0.014}$ | $0.346_{\pm 0.019}$ | $0.429_{\pm 0.039}$ | $0.477_{\pm 0.039}$ |
| | 0.05 | $0.288_{\pm 0.019}$ | $\underline{0.321_{\pm 0.018}}$ | $\mathbf{0.378_{\pm 0.025}}$ | $\mathbf{0.406_{\pm 0.042}}$ |
| | 0.1 | $0.287_{\pm 0.006}$ | $\mathbf{0.315_{\pm 0.017}}$ | $\underline{0.388_{\pm 0.029}}$ | $0.437_{\pm 0.038}$ |
| | 0.2 | $\mathbf{0.283_{\pm 0.010}}$ | $0.332_{\pm 0.019}$ | $0.432_{\pm 0.026}$ | $0.495_{\pm 0.042}$ |
| | 0.5 | $\underline{0.286_{\pm 0.014}}$ | $0.366_{\pm 0.025}$ | $0.533_{\pm 0.040}$ | $0.653_{\pm 0.057}$ |

**QP Smoothing.** We sweep $\lambda \in \{0.01, 0.05, 0.1, 0.5, 1.0\}$ with $\beta = 0.1$ fixed. Small $\lambda$ keeps the forward solution close to the original LP but yields nearly singular projections; large $\lambda$ stabilizes the projection but biases the forward solution away from the LP optimum. Table 9 reports the normalized regret. On Shortest Path, $\lambda = 0.1$ is optimal at every degree, while $\lambda = 0.5$ is preferred on Knapsack. Across the grid, $\lambda \in [0.1, 0.5]$ stays close to the best regret, and we use $\lambda = 0.1$ as the default.

*Table 9.* **QP-smoothing ablation.** Normalized regret (%, ↓) under varying $\lambda$, with $\beta = 0.1$ fixed. Best $\lambda$ per degree in **bold**, second best underlined.

| Task | $\lambda$ | Deg 2 | Deg 4 | Deg 6 | Deg 8 |
|---|---|---|---|---|---|
| Shortest Path | 0.01 | $0.471_{\pm 0.281}$ | $1.834_{\pm 0.803}$ | $4.554_{\pm 1.838}$ | $9.410_{\pm 3.792}$ |
| | 0.05 | $0.152_{\pm 0.049}$ | $0.898_{\pm 0.301}$ | $2.421_{\pm 0.859}$ | $5.906_{\pm 2.529}$ |
| | 0.1 | $\mathbf{0.104_{\pm 0.034}}$ | $\mathbf{0.774_{\pm 0.239}}$ | $\mathbf{2.138_{\pm 0.681}}$ | $\mathbf{4.246_{\pm 0.989}}$ |
| | 0.5 | $\underline{0.114_{\pm 0.044}}$ | $1.221_{\pm 0.325}$ | $3.078_{\pm 0.964}$ | $\underline{4.952_{\pm 1.506}}$ |
| | 1.0 | $0.128_{\pm 0.029}$ | $1.384_{\pm 0.329}$ | $3.992_{\pm 1.224}$ | $6.878_{\pm 2.462}$ |
| Knapsack | 0.01 | $0.361_{\pm 0.007}$ | $0.386_{\pm 0.024}$ | $0.521_{\pm 0.021}$ | $0.709_{\pm 0.110}$ |
| | 0.05 | $0.293_{\pm 0.011}$ | $0.329_{\pm 0.021}$ | $0.419_{\pm 0.025}$ | $0.473_{\pm 0.045}$ |
| | 0.1 | $0.287_{\pm 0.006}$ | $\underline{0.315_{\pm 0.017}}$ | $0.388_{\pm 0.029}$ | $0.437_{\pm 0.038}$ |
| | 0.5 | $\mathbf{0.268_{\pm 0.014}}$ | $\mathbf{0.311_{\pm 0.019}}$ | $\mathbf{0.370_{\pm 0.027}}$ | $\mathbf{0.391_{\pm 0.029}}$ |
| | 1.0 | $\underline{0.269_{\pm 0.010}}$ | $0.322_{\pm 0.021}$ | $0.409_{\pm 0.029}$ | $\underline{0.429_{\pm 0.033}}$ |

## G.2. Real-World Knapsack: Building Investment

Real-world knapsack benchmarks are uncommon in the DFL literature, where the standard practice is to use synthetic feature–cost mappings as in our LP benchmarks above. For completeness, we additionally report results on the building investment dataset of Mandi & Guns (2020), one of the few real-world knapsack instances used in prior DFL work, with 372 properties labeled by construction cost and sales price.

**Problem Formulation.** Each test instance solves a binary knapsack with predicted sales prices as item values and known construction costs as weights.

$$
\begin{aligned}
\max_{z} \quad & \sum_{i=1}^{n} \hat{c}_i z_i \\
\text{subject to} \quad & \sum_{i=1}^{n} w_i z_i \leq B \\
& z_i \in \{0, 1\}, \quad \forall i \in \{1, \ldots, n\}
\end{aligned}
\tag{46}
$$

Here $\hat{c}_i$ is the predicted sales price of property $i$, $w_i$ is its construction cost, and $B$ is the budget. We follow the setup of Mandi & Guns (2020) with two changes. We use an inequality budget constraint instead of equality, which avoids

infeasibility across batch sizes, and we use a batch size of 10 instead of 31. The budget is set to $B = 0.4 \sum_i w_i$, all other settings follow Appendix E.1, and we report normalized regret over 5 seeds.

*Table 10.* **Real-world Knapsack (building investment).** Normalized regret (%, ↓) over 5 seeds. Best in **bold**, second best underlined.

| Method | MSE | SPO+ | PFYL | DBB | LAVA | PEAR |
|---|---|---|---|---|---|---|
| Regret (%) | $1.22_{\pm 0.89}$ | $0.72_{\pm 0.57}$ | $\underline{0.70}_{\pm 0.80}$ | $12.20_{\pm 10.05}$ | $2.12_{\pm 0.79}$ | $\mathbf{0.51}_{\pm \mathbf{0.57}}$ |

Table 10 reports the normalized regret. PEAR achieves the lowest regret, with PFYL second. The two-stage MSE baseline trails the leading DFL methods, and DBB shows the largest regret with substantial variance.

## H. Additional Discussion

**Stability of Active-Set Identification.** Our theoretical results rely on correctly identifying the active constraint set at the optimal solution, which in practice is inferred from a numerical solver and can be sensitive near constraint boundaries; thus constraints that are close to active can be misclassified (Oberlin & Wright, 2006). When this happens, the reduced Jacobian $J$ used in the projection is perturbed, which can in turn perturb the projected gradient.

To quantify this effect, we measure how often the inferred active set changes under small perturbations. Given an optimization problem with $m$ inequality constraints, we encode the activity pattern as a binary mask $a \in \{0,1\}^m$. For a reference solution with mask $a$ and a perturbed solution with mask $a'$, we report the change rate in percent

$$\rho\,(\%) \;=\; \frac{100}{m} \sum_{i=1}^{m} \mathbb{I}[a_i \neq a_i']. \tag{47}$$

We generate multiple perturbed costs around $\hat{c}$, recompute the corresponding masks, and average $\rho$ across perturbations. In our LP benchmarks, the average change rates were $\rho = 0.30\%$ for Shortest Path and $\rho = 0.14\%$ for Knapsack. These results validate the stability of active-set identification in our experiments. For problems with many nearly binding constraints, the gradient estimate may be less stable.

*Table 11.* **Prediction accuracy.** MSE (↓), over 5 seeds. Best in **bold** and second best underlined.

(a) LP Benchmarks

| Task | Method | Deg 2 | Deg 4 | Deg 6 | Deg 8 |
|---|---|---|---|---|---|
| Shortest Path | MSE | $\mathbf{0.00}_{\pm \mathbf{0.00}}$ | $\mathbf{0.09}_{\pm \mathbf{0.01}}$ | $\mathbf{0.68}_{\pm \mathbf{0.08}}$ | $\mathbf{4.59}_{\pm \mathbf{1.08}}$ |
| | SPO+ | $0.31_{\pm 0.18}$ | $0.61_{\pm 0.23}$ | $1.55_{\pm 0.17}$ | $6.24_{\pm 1.21}$ |
| | PFYL | $1.78_{\pm 0.06}$ | $1.53_{\pm 0.07}$ | $1.86_{\pm 0.05}$ | $5.64_{\pm 1.17}$ |
| | DBB | $0.10_{\pm 0.04}$ | $0.20_{\pm 0.09}$ | $1.14_{\pm 0.26}$ | $5.65_{\pm 1.02}$ |
| | LAVA | $1.16_{\pm 0.04}$ | $1.19_{\pm 0.05}$ | $2.37_{\pm 0.23}$ | $7.89_{\pm 1.21}$ |
| | PEAR | $\underline{0.01}_{\pm 0.01}$ | $\underline{0.16}_{\pm 0.06}$ | $\underline{0.78}_{\pm 0.09}$ | $\underline{5.23}_{\pm 1.18}$ |
| Knapsack | MSE | $\mathbf{0.18}_{\pm \mathbf{0.01}}$ | $\mathbf{2.67}_{\pm \mathbf{0.26}}$ | $\mathbf{22.58}_{\pm \mathbf{3.18}}$ | $\mathbf{139.42}_{\pm \mathbf{21.60}}$ |
| | SPO+ | $4.59_{\pm 0.09}$ | $5.82_{\pm 0.35}$ | $25.14_{\pm 3.10}$ | $141.33_{\pm 24.10}$ |
| | PFYL | $8.97_{\pm 0.11}$ | $8.96_{\pm 1.52}$ | $36.16_{\pm 5.81}$ | $181.77_{\pm 27.89}$ |
| | DBB | $12.07_{\pm 0.24}$ | $11.89_{\pm 0.63}$ | $29.59_{\pm 4.80}$ | $144.13_{\pm 24.16}$ |
| | LAVA | $10.97_{\pm 0.60}$ | $16.27_{\pm 1.66}$ | $50.65_{\pm 5.63}$ | $205.00_{\pm 31.15}$ |
| | PEAR | $\underline{2.54}_{\pm 2.53}$ | $12.95_{\pm 2.01}$ | $43.34_{\pm 4.96}$ | $187.62_{\pm 29.71}$ |

(b) Real-world QP

| Method | Normalized MSE |
|---|---|
| MSE | $\mathbf{1.105 \pm 0.005}$ |
| PEAR | $\underline{1.457 \pm 0.016}$ |
| QPTH | $1.469 \pm 0.024$ |
| CvxpyLayers | $1.493 \pm 0.019$ |

**Prediction Accuracy and Decision Quality.** Existing DFL methods focus on minimizing regret, often at the expense of prediction accuracy measured by MSE. Our framework explains this phenomenon by decomposing prediction error into a decision-relevant tangent component and a decision-irrelevant normal component. Conventional DFL methods implicitly discard the normal component, which can degrade prediction accuracy.

This relationship between regret minimization and prediction accuracy is highly problem-dependent. Since PEAR prioritizes regret reduction, it adaptively allocates learning capacity based on the problem structure. In Shortest Path, where accurate cost prediction of neighboring nodes is inherently linked to identifying optimal paths, PEAR achieves substantially lower

MSE compared to other DFL baselines. In contrast, for Knapsack where relative ranking of item values matters more than precise magnitude estimation, PEAR focuses more on decision quality improvement rather than prediction accuracy.

Our constraint shift experiments further support this observation. In Shortest Path and Portfolio Optimization, where the MSE baseline achieved the best regret under constraint shift, PEAR also exhibited superior prediction accuracy compared to other DFL methods. However, in Knapsack where the MSE baseline suffered significant regret degradation under constraint shifts, PEAR showed higher MSE than other DFL baselines. These results suggest that PEAR implicitly learns to maintain prediction fidelity when it is beneficial for decision quality.

**Limitations and Future Directions.**  While existing DFL methods have achieved strong decision quality, they have operated as black boxes with respect to why regret decreases and how the learning direction is determined. Our work provides a structural characterization that makes these dynamics interpretable through the lens of tangent space projection. Nevertheless, PEAR remains within the DFL paradigm and thus shares its fundamental limitation: an inherent bias toward decision quality at the potential expense of prediction accuracy.

Through our theoretical analysis, we established that the regret gradient lies within the MSE gradient as its projection onto the decision-relevant subspace. The Table 11 results demonstrate that learning only the decision-relevant component can still achieve strong prediction accuracy when problem structure tightly couples prediction and decision quality. Leveraging this decomposition into normal and decision-relevant components to overcome the trade-off between decision quality and prediction accuracy, long viewed as a structural limitation of DFL, remains a promising direction for future work.

