# OpenReview forum: "Decision-Focused Learning via Tangent-Space Projection of Prediction Error"
_ICML.cc/2026/Conference — ICML 2026 regular_

### Official Review · Reviewer_tprh · 2026-02-19

**Soundness:** 3
**Presentation:** 3
**Significance:** 2
**Originality:** 3
**Overall Recommendation:** 4
**Confidence:** 3

**Summary:**

To address the high computational cost or bias in computing regret gradients in Decision-Focused Learning (DFL), the authors propose the PEAR algorithm. With standard regularity conditions and locally stable active constraints, the regret gradient has a closed-form geometric characterization. It computes regret gradients by solving a reduced linear system constructed only over active constraints. Experiments on several tasks show that PEAR outperforms all baseline methods in decision quality while achieving the highest computational efficiency, with advantages persisting under constraint shifts.

**Compliance With Llm Reviewing Policy:**

Affirmed.

**Final Justification:**

Agree with the acception.

**Key Questions For Authors:**

1. Computational time is not the fastest among the compared baselines. How to say the efficiency of the proposed method is the best?
2. The numerical experiments do not examine the algorithm's performance under varying problem scales, numbers of active constraints, or frequencies of constraint changes.
3. The results only compare the regret and time. Figure for iteration comparation is not provided.

**Limitations:**

1. Regarding computational efficiency, the algorithm's runtime is not the fastest among all baselines.
2. Efficiency should be evaluated holistically, considering metrics such as time-to-solution for a given regret target or iteration complexity, more than relying solely on end-to-end runtime.
3. The theoretical analysis and experiments are confined to problems with convex or LP/QP structures. The performance and potential challenges of extending the method to non-convex decision problems remain unexplored, which limits the perceived generality of the approach. Addressing these points would strengthen the empirical foundation and clarify the method's practical applicability.

**Strengths And Weaknesses:**

Strengths​:
The paper offers a valuable theoretical insight by revealing the geometric nature of regret gradients and introduces a conceptually elegant algorithm that avoids costly differentiation through solver iterations. The closed-form characterization of regret gradients under stable active constraints provides a clear and intuitive interpretation, advancing the theoretical understanding of Decision-Focused Learning. The method is systematically derived and experimentally validated on canonical LP/QP tasks, demonstrating its potential in structured convex settings.

Weaknesses​:
1. Although computational efficiency is highlighted, the reported runtime is not the fastest among all baselines, and efficiency is assessed narrowly via end-to-end time rather than more informative measures such as time-to-solution for fixed regret targets or iteration-wise convergence speed.
2. Experiments are restricted to moderate-scale and convex problems, with no analysis of how performance scales with problem size, varying numbers of active constraints, or dynamic constraint changes.
3. Results focus only on final regret and total runtime, omitting iteration-level convergence plots that would help diagnose optimization behavior.

---

> ### Author Rebuttal · Authors · 2026-03-28
>
> We thank the reviewer for the careful review and constructive questions.
>
> ---
>
> **Q1. Computational efficiency of PEAR.**
>
> From Tables 1 and 3, PEAR is the fastest among the DFL baselines except for the two-stage MSE baseline. Since MSE minimizes prediction error rather than regret, our main comparison is among DFL methods, where PEAR shows the strongest efficiency.
>
> As noted in the review, there is one exception in KS with degree 8, where LAVA is faster than PEAR. However, this occurs only in a single case, so our overall claim remains valid. More importantly, even in that setting, PEAR achieves much better regret (0.437%) than LAVA (1.351%), showing the strongest performance-efficiency trade-off among the DFL baselines.
>
> ---
>
> **Q2. Additional results with problem size and active constraints.**
>
> As in most DFL work, the goal of this paper is not large-scale optimization itself, but understanding how regret-based learning signals can be used to train predictive models for better downstream tasks. Existing DFL papers also typically study problems with tens to hundreds of decision variables. PyEPO’s SP, KS, and TSP have 40, 32, and 190 decision variables, respectively, and [1,2] also consider similar scales. In this sense, our settings are already in line with standard DFL benchmarks.
>
> Nonetheless, we appreciate the reviewer's concern regarding scalability and so conducted additional experiments on the hardest degree 8 settings by roughly doubling the decision variables. In SP, the number of edges increased from 40 to 84 and the number of flow-conservation constraints from 25 to 49. In KS, the number of items increased from 100 to 200. As shown in the table, PEAR still achieved the best regret in both cases, and in SP it also remained the fastest.
>
> **SP**
>
> | Method | Regret Mean ± Std | Time Mean ± Std |
> | --- | --- | --- |
> | MSE | 16.41 ± 3.61 | 4.01 ± 0.90 |
> | SPO | 5.05 ± 1.82 | 56.64 ± 24.07 |
> | DBB | 12.41 ± 3.71 | 87.97 ± 25.36 |
> | PFYL | 5.32 ± 1.49 | 46.22 ± 12.72 |
> | LAVA | 6.52 ± 1.95 | 135.17 ± 8.80 |
> | **PEAR** | **4.34 ± 1.18** | **39.34 ± 20.47** |
>
> **KS**
>
> | Method | Regret Mean ± Std | Time Mean ± Std |
> | --- | --- | --- |
> | MSE | 1.98 ± 0.29 | 129.17 ± 12.08 |
> | SPO | 0.74 ± 0.05 | 600.00 ± 0.00 |
> | DBB | 0.92 ± 0.07 | 600.00 ± 0.00 |
> | PFYL | 0.99 ± 0.14 | 600.00 ± 0.00 |
> | LAVA | 1.50 ± 0.26 | 235.67 ± 85.75 |
> | **PEAR** | **0.40 ± 0.02** | **444.04 ± 146.70** |
>
> The number of active constraints is sample-dependent and varies across problem types. In shortest path problems, we expect it to increase with the number of edges, since optimal solutions lie on vertices where many bound constraints are active. Despite this growth, PEAR remains stable and trains effectively even when the active set becomes large.
>
> More importantly, Appendix G demonstrates that active-set identification is highly stable: under cost perturbations, the change rate is only 0.30% for SP and 0.14% for KS, indicating that the active-set structure remains largely consistent across samples.
>
> ---
>
> **Q3. Lack of iteration-level convergence comparisons.**
>
> We report test regret and training time, but not iteration-level convergence plots, due to our training setup. We use validation-based early stopping, so the reported training time reflects the wall-clock time until convergence.
>
> Under this protocol, shorter training time directly implies faster convergence. Since PEAR achieves the best regret with the shortest training time in almost all settings, we believe convergence behavior is reflected in the results.
>
> In addition, iteration counts are not directly comparable across methods because their per-iteration costs differ (PEAR: reduced system; SPO+: solver call; DBB: perturbation sampling). While iteration-wise plots may provide additional diagnostics, we believe that regret and training time are more direct and practical measures of efficiency.
>
> ---
>
> **Q4. Generality beyond convex problems.**
>
> Our regret gradient analysis and training procedure are developed for convex problems. In LP experiments, curvature is injected to obtain meaningful regret gradients during training. However, this does not mean that our results are limited to convex problems.
>
> KS, one of our main benchmarks, is originally a 0–1 integer program and thus a combinatorial NP-hard problem. During training, we use LP relaxation and smoothing to obtain informative learning signals, but evaluation is performed on the original integer problem.
>
> Therefore, our point is not that PEAR provides a general theory for arbitrary non-convex optimization, but that structured convex regret gradient signals can still improve decision quality in non-convex problems.
>
> ---
>
> We hope this clarifies our intent, and we will revise the manuscript accordingly.
>
> **Ref.**
>
> [1] Shah et al. (2022), *Decision-Focused Learning without Differentiable Optimization*, NeurIPS.
>
> [2] Shah et al. (2024), *Leaving the Nest: Going Beyond Local Loss Functions for Predict-Then-Optimize*, AAAI.

---

> > ### Author Rebuttal · Reviewer_tprh · 2026-04-01
> >
> > No other questions.

---

> > > ### Author Response · Authors · 2026-04-01
> > >
> > > Thank you for the acknowledgement and for the careful consideration of our rebuttal. We are glad that the concerns have been adequately addressed. If the clarification and additional results are sufficient, we would be grateful if this could be reflected in your final assessment.

---

### Official Review · Reviewer_p13U · 2026-03-12

**Soundness:** 3
**Presentation:** 3
**Significance:** 3
**Originality:** 3
**Overall Recommendation:** 5
**Confidence:** 4

**Summary:**

The paper proposes an efficient method called PEAR for computing regret gradients in a DFL setting. The method computes regret gradients via a reduced linear system over active constraints, rather than differentiating through iterative optimization solver calls, which improves training efficiency. Most interestingly, the authors show that the regret gradient can be obtained from the MSE gradient by filtering out decision-irrelevant components of the prediction error.

**Compliance With Llm Reviewing Policy:**

Affirmed.

**Final Justification:**

I'd like to thank the authors for the detailed response and adding new results, and sorry for the late reply. The explanation of PEAR’s training time is clear, and the new real‑world evaluation is good. I’ve raised my score.

**Key Questions For Authors:**

1.Can the core observation, the regret gradient can be obtained from the MSE gradient by filtering out decision-irrelevant components of the prediction error, be applied to scenarios that unknowns also in constraints?

2.The authors claimed that the proposed method can achieve much shorter training times that others, but in “Knapsack” in Table 1, the training time of proposed PEAR is almost the same as others when Deg 2, and even larger than that of LAVA when Deg 8. Could the authors explain why?

**Limitations:**

See “Weaknesses”.

**Strengths And Weaknesses:**

Strengths:

1. The paper offers a new perspective on gradient computation for DFL, which is interesting and efficient.

2. The writing is generally clear.

3. The application scope of the proposed method is broad, covering LP and QP.

Weaknesses:

1. Most experiments are conducted on simulated datasets. However, realistic datasets for KP, for example, are available and widely used in DFL research. Why not evaluate the proposed method on real-world datasets?

2. The work assumes that the unknown parameters appear only in the objective. Cases where unknowns appear in the constraints require further investigation.

---

> ### Author Rebuttal · Authors · 2026-03-28
>
> We thank the reviewer for the careful and thoughtful review of our paper. We also appreciate the positive assessment, and we address the concerns and questions raised in the review.
>
> ---
>
> **Q1. PEAR with uncertainty in the constraints.**
>
> Thank you for suggesting this possible extension. However, uncertainty in the constraints is fundamentally different from the setting considered in our work and in most DFL papers. Most existing work focuses on uncertainty in the objective and our paper follows this setup. In that setting, the goal is to learn predictions that improve downstream decision quality. By contrast, when unknown parameters appear in the constraints, the main issue is how accurately the feasible set is estimated so that feasibility is preserved. Thus, objective uncertainty and constraint uncertainty have different structures and are not naturally handled in the same framework.
>
> Our method interprets regret gradients in the standard DFL setting through the local geometry of the active constraints. Extending this analysis to uncertain constraints is therefore not straightforward. This setting has recently begun to be studied as a separate problem class, with feasibility-aware DFL methods proposed specifically for it [1]. While interesting, it is outside the scope of the current paper.
>
> ---
>
> **Q2. PEAR’s training time.**
>
> We measured training time under early stopping. It therefore reflects end-to-end training efficiency. As you noted, KS with degree 8 is a case where PEAR takes longer than expected. However, this occurs only in a single case, so our overall claim about efficiency remains valid.
>
> The behavior at degree 2 is expected. When the degree is low, the feature-cost relation is close to linear. As a result, even MSE achieves relatively low regret, and all methods converge quickly. Differences become clearer as the degree increases and the problem becomes more nonlinear; the performance gap between PEAR and other methods is largest at degree 8.
>
> More importantly, PEAR shows the strongest **performance-efficiency trade-off** among the DFL baselines. In KS degree 8, LAVA is faster, but PEAR achieves **0.437%** regret, compared to 1.351% for LAVA. By contrast, PEAR uses a structured regret gradient while keeping backward computation efficient through the reduced system. Therefore, efficiency remains one of the main contributions of our work.
>
> ---
>
> **Q3. LP benchmarks and evaluation on real-world data.**
>
> Our experiments follow the standard settings most widely used in the DFL literature. For the LP tasks, we use the PyEPO benchmark framework [2], and evaluate SP and KS in settings that allow direct comparison with prior DFL work [3,4,5].
>
> While some DFL studies have used real-world feature sources for KS experiments [6,7], these setups vary across papers and no single real-world benchmark has become standard in the field. We therefore follow the widely-used PyEPO-based benchmarks [2], which allow direct and fair comparison with representative baselines under matched conditions. We further broaden the evaluation through degree, noise, and constraint-shift experiments.
>
> While the LP sections use synthetic data, the QP section evaluates PEAR in a real-world setting. Under mean-variance portfolio optimization, we use real S&P 100 market data from 2010 to 2024, and PEAR achieves the best results. The overall message is therefore not that PEAR works only on synthetic data, but that it is effective both on standard DFL benchmarks and on a real-world task built from real financial data.
>
> ---
>
> To the best of our knowledge, this is the first work in DFL to study the structure of the regret gradient in a fundamental way. The paper is also supported by a broad empirical evaluation, including LP/QP settings, noise, and constraint shifts, and it explicitly reports training time, which has rarely been examined in prior DFL work. Together, these results provide a careful validation of both the proposed regret gradient analysis and the practical effectiveness of the method.
>
> We hope this clarifies the scope of our work and the rationale behind our experimental design.
>
> **Ref.**
>
> [1] Mandi et al. (2025), *Feasibility-Aware Decision-Focused Learning for Predicting Parameters in the Constraints*, NeurIPS.
>
> [2] Tang and Khalil (2022), *PyEPO: A PyTorch-Based End-to-End Predict-Then-Optimize Library for Linear and Integer Programming*, arXiv.
>
> [3] Schutte et al. (2024), *Robust Losses for Decision-Focused Learning*, IJCAI.
>
> [4] Gupta and Huang (2024), *Decision-Focused Learning with Directional Gradients*, NeurIPS.
>
> [5] Berden et al. (2025), *Solver-Free Decision-Focused Learning for Linear Optimization Problems*, NeurIPS.
>
> [6] Mulamba et al. (2020), *Contrastive Losses and Solution Caching for Predict-and-Optimize*, IJCAI.
>
> [7] Mandi and Guns (2020), *Interior Point Solving for LP-Based Prediction+Optimisation*, NeurIPS.

---

> > ### Author Rebuttal · Reviewer_p13U · 2026-04-03
> >
> > Thank you for the detailed response. The explanation of PEAR’s training time is clear, but the lack of real‑world evaluation remains concerning. I’ll keep my score.

---

> > > ### Author Response · Authors · 2026-04-04
> > >
> > > Thank you for acknowledging our training time explanation and for the continued engagement. We address the remaining concern below.
> > >
> > > **1. The LP benchmarks follow the dominant standard in DFL.**
> > >
> > > The SPO+/PyEPO framework is the most widely adopted and standardized evaluation protocol for LP-based DFL. Numerous works follow this framework or its data generation logic to benchmark their methods, including [1,2,3,4,5,6] among others. Our LP experiments follow this established protocol precisely to enable direct and fair comparison across representative baselines.
> > >
> > > The reviewer mentioned that real-world KS datasets are “available and widely used in DFL research.” However, we believe real-world KS benchmarks remain relatively limited in the current DFL literature. Only a few papers have used real-world feature sources for KS experiments [7,8], and even in these cases, the knapsack weights are randomly generated rather than obtained from real data.
> > >
> > > **2. The QP evaluation uses carefully designed real-world data.**
> > >
> > > For portfolio optimization (MVO), we use real S\&P 100 market data from 2010 to 2024. Our experimental setup incorporates practical considerations such as rebalancing periods and transaction costs, closely reflecting real-world portfolio management decisions. To our knowledge, very few DFL studies address MVO with this level of detail. Because most existing DFL surrogate losses are restricted to linear objectives, few prior works address QP settings directly. Among those that do, each uses a different experimental setup. Our MVO experiment was designed with particular care, and PEAR achieves the best results in this setting.
> > >
> > > **3. The theoretical contribution is independent of data source.**
> > >
> > > The core result of this paper, that the regret gradient is the projection of the MSE gradient onto the active constraint tangent space, is a structural property of LP/QP with uncertain objectives. This insight holds regardless of whether the data is synthetic or real. The experiments serve to validate this theoretical result under controlled conditions (varying degree, noise, and constraint shifts) and to demonstrate practical effectiveness.
> > >
> > > **4. Additional real-world KS experiment.**
> > >
> > > Nevertheless, to directly address the reviewer's concern, we conducted an additional experiment on a real-world knapsack dataset. Among the few real-world KS experiments in DFL, we selected the building investment dataset from [9] because, unlike other setups where knapsack weights are randomly generated, [9] uses actual construction costs as item weights, making it the most realistic setting available. The dataset contains 372 properties with construction costs and sales prices from the real estate market. Each test instance solves the following binary knapsack problem:
> > >
> > > $\max_{\mathbf{x} \in \{0,1\}^n} \hat{\mathbf{c}}^\top \mathbf{x} \quad \text{s.t.} \quad \mathbf{w}^\top \mathbf{x} \leq B$
> > >
> > > where $\hat{\mathbf{c}}$ is the predicted sales price, $\mathbf{w}$ is the known construction cost, and $B$ is the budget constraint. We use the same dataset, model architecture, and train/test split as [9], with two modifications: (i) an inequality constraint ($\leq$) instead of the equality constraint in [9], as the latter requires spending the budget exactly and causes infeasibility when batch size changes; (ii) a batch size of 10 instead of 31, yielding 6 test batches for more reliable evaluation. We set $B = 0.4 \times \sum_i w_i$ to ensure feasibility regardless of batch size.
> > >
> > > | Method | Regret (\%) |
> > > |--------|-----------|
> > > | PEAR | **0.51 ± 0.57** |
> > > | PFYL | 0.70 ± 0.80 |
> > > | SPO+ | 0.72 ± 0.57 |
> > > | MSE | 1.22 ± 0.89 |
> > > | LAVA | 2.12 ± 0.79 |
> > > | DBB | 12.20 ± 10.05 |
> > >
> > > PEAR achieves the lowest regret among all methods. We will incorporate these results into the revised manuscript. We hope the above thoroughly addresses the reviewer's concern regarding real-world evaluation, and we would appreciate it if they could be considered in the final assessment.
> > >
> > > **Ref.**
> > >
> > > [1] Elmachtoub and Grigas (2022), Smart "Predict, then Optimize", Management Science.
> > >
> > > [2] Mandi et al. (2022), Decision-Focused Learning: Through the Lens of Learning to Rank, ICML.
> > >
> > > [3] Schutte et al. (2024), Robust Losses for Decision-Focused Learning, IJCAI.
> > >
> > > [4] Gupta and Huang (2024), Decision-Focused Learning with Directional Gradients, NeurIPS.
> > >
> > > [5] Tang and Khalil (2024), CaVE: A Cone-Aligned Approach for Fast Predict-then-Optimize, CPAIOR.
> > >
> > > [6] Chen et al. (2023), Landscape Surrogate: Learning Decision Losses for Mathematical Optimization Under Partial Information, NeurIPS.
> > >
> > > [7] Mulamba et al. (2021), Contrastive Losses and Solution Caching for Predict-and-Optimize, IJCAI.
> > >
> > > [8] Berden et al. (2025), Solver-Free Decision-Focused Learning for Linear Optimization Problems, NeurIPS.
> > >
> > > [9] Mandi and Guns (2020), Interior Point Solving for LP-based Prediction+Optimisation, NeurIPS.

---

### Official Review · Reviewer_S74w · 2026-03-12

**Soundness:** 3
**Presentation:** 3
**Significance:** 3
**Originality:** 2
**Overall Recommendation:** 5
**Confidence:** 4

**Summary:**

This paper investigates the structure of regret gradients in decision-focused learning and whether this structure can be directly leveraged to simplify the training process. Considering convex optimization problems with linear constraints and linear or quadratic objectives, the authors formulate the regret gradient as a projection of the prediction error onto the tangent space after scaling by local curvature. Building upon this result, the paper proposes PEAR, which computes gradients using active-set reduction and a reduced Schur-complement system. Experiments cover two LP benchmarks, a mean-variance portfolio QP, and scenarios involving noisy perturbations and constraint shifts.

**Compliance With Llm Reviewing Policy:**

Affirmed.

**Final Justification:**

No further questions.

**Key Questions For Authors:**

1. Compared to classical sensitivity analysis, active-set reduction, and the geometric formulations already presented in Gould et al. (2021), what exactly is the new theoretical contribution introduced in this paper? Could the authors separately outline the standard results, the reorganized interpretation, and the novel aspects of this paper?

2. Since the main theorem requires `H ≻ 0`, and the primary experiments focus on LP benchmarks, I am particularly interested in understanding the distinct roles of quadratic smoothing and `normal injection`. Specifically, how much relationship remains between the updated direction and the original regret gradient after incorporating the normal component?

3. Since the paper's core conclusion is that regret gradients can be derived from projected prediction error, could the authors directly compare PEAR gradients with exact implicit regret gradients in computable QP scenarios?

4. Once constraint shift enters the testing phase, evaluation becomes closer to generalization after feasible region changes. Could the authors more explicitly clarify the relationship between these experiments and the main theory? Furthermore, when `MSE` outperforms PEAR in several shift scenarios, how should we interpret the advantages of the proposed method?

5. The paper repeatedly emphasizes computational efficiency, but based on the implementation description, PEAR still requires forward solving, active-set detection, and reduced linear solving. Therefore, I would like to see a more detailed runtime breakdown and whether the method retains its advantages when the active set grows larger or becomes more unstable.

**Limitations:**

The author has addressed some constraints, but I still recommend formulating the boundary conditions more explicitly. This is because the theoretical results in this paper rely on the local active set being fixed, strict complementarity, and `H ≻ 0`. Meanwhile, the main LP experiments require smoothing and injection to run. Consequently, the strongest theoretical conclusions do not directly cover the most critical experimental section of the paper. Additionally, the constraint shift evaluation assesses generalization after feasible region shifts, which requires separate clarification in the main text.

**Strengths And Weaknesses:**

### Strengths
- The paper analyzes the structure of the regret gradient and directly links this structure to the construction of training algorithms. This approach is clear and differs from methods that solely emphasize differentiable solvers.
- Under the conditions of a fixed local active set and `H ≻ 0`, the paper provides an intuitive explanation through tangent and normal decompositions, enabling readers to understand which error components influence downstream decisions.
- Experiments cover LP, QP, noisy perturbations, and constraint variations. While not extensive, they at least provide results across several distinct scenarios.

### Weaknesses
- The paper fails to clearly articulate its relationship with prior work. While the main text acknowledges that Gould et al. (2021) presented a similar geometric decomposition, the core derivation heavily relies on sensitivity analysis and active-set reduction. Consequently, readers remain uncertain about the novel theoretical contributions of this paper.
- The main theorem holds only under the conditions of a locally fixed active set, strict complementarity, and `H ≻ 0`. However, the main text fails to quantify the stability of these conditions during training and does not directly link active-set identification errors to final performance.
- There is a noticeable disconnect between the LP section and the main theory, as the original LP does not satisfy `H ≻ 0`. The authors subsequently introduce quadratic smoothing and `normal injection` to obtain the training signal, which in turn raises doubts about whether the update direction still corresponds to the original regret gradient.
- Experimental results are insufficient to directly support the geometric interpretation, as the main paper neither compares `P_H e` with exact implicit regret gradients nor provides systematic ablation studies for `normal injection`, `beta`, and `lambda`.
- The conclusions in the abstract and main text are overstated relative to the results. While `SPO+` exhibits lower regret under certain settings and `MSE` demonstrates greater robustness in some constraint shift scenarios, it is more accurate to state that PEAR performs strongly against select DFL baselines.
- Constraint shift evaluates generalization after feasible region changes, addressing a different level of problem than the main theorem. The current text fails to clarify this distinction.

---

> ### Author Rebuttal · Authors · 2026-03-28
>
> We sincerely thank the reviewer for the thoughtful and detailed review. We appreciate the valuable comments.
>
> ---
> **Q1. Theoretical contribution beyond classical sensitivity analysis.**
>
> KKT sensitivity analysis and active-set reduction are classical [2,3]. While [1] studies optimization-gradient geometry, it does not address DFL, regret gradients, or their link to prediction error. Our contribution is to make this connection clear. Theorem 3.5 shows that the regret gradient is the tangent-space projection of the prediction error, i.e., the MSE gradient with irrelevant components removed. This interpretation is absent in [1] or prior DFL work. Prior DFL methods focus on **'How'** to obtain regret gradient, e.g., via solver differentiation, surrogate losses, or perturbations. In contrast, we first ask **'What'** the regret gradient structurally is, and PEAR follows directly from that characterization.
>
> ---
> **Q2. Distinct roles of smoothing and normal injection.**
>
> QP smoothing adds a small quadratic term to the LP objective, following [4]. Since evaluation is on the original LP, improved decision quality shows the smoothed QP gives a useful signal for the LP.
>
> Normal injection is optional. As [7] shows, regret gradients can still vanish after smoothing. To address the reviewer’s question, we ran additional ablation experiments; the results are available at https://anonymous.4open.science/r/ablation-results-22EF. Even with $\beta=0$, performance remains competitive, and in some settings it even outperforms our default choice. We therefore fix $\beta$ and $\lambda$ to robust values that work well across settings.
>
> ---
> **Q3. Exact gradient comparison in QP settings.**
>
> PEAR and differentiable layers compute the same gradient in theory. The main difference is numerical stability. QPTH can suffer from ill-conditioning due to symmetrization, and CVXPYLayers has limitations from cone reformulation [5]. PEAR avoids much of this through the reduced system.
>
> We additionally compare gradients in QP. Using an untrained DLinear model, we compared PEAR gradients with finite-difference ground truth on the S&P 100 test set. Each dimension was perturbed by 1e-5, the QP was repeatedly solved with Gurobi, and regret changes were measured. Over all 42 batches, cosine similarity was >0.90 on every batch and >0.99 on 41 batches, showing PEAR accurately computes the exact regret gradient.
>
> ---
> **Q4. Interpretation of constraint-shift experiments.**
>
> We agree that constraint shift is not a direct verification of the theorem, but a generalization test. Its purpose is to show some DFL methods over-specialize to training decisions. In SP, changing the source and destination diagonally causes the regret of SPO+, PFYL, and LAVA to increase sharply, whereas PEAR achieves the lowest regret among DFL methods.
>
> PEAR adapts to problem structure. In SP, neighboring edge costs directly affect route selection, so accurate cost prediction leads to better decisions. Table 8 in Appendix G shows that PEAR has the highest prediction accuracy among DFL methods. In KS, relative ranking matters more than absolute values, so PEAR prioritizes decision quality over prediction accuracy and achieves the lowest regret in both settings. MSE can appear robust under SP shift because it learns all costs uniformly, but this is also why it performs worst in the original setting. Across shifts, PEAR consistently maintains the lowest regret among DFL methods.
>
> ---
> **Q5. Runtime breakdown and active-set behavior.**
>
> The forward solve is shared across DFL methods, and active-set detection adds no extra optimization because it is obtained from the primal-dual solution returned by OSQP [6]. The reduced linear solve replaces the full KKT solve used by differentiable layers, so the backward pass is simpler. Table 1 shows that PEAR reaches the best regret within the shortest training time in nearly all settings.
>
> For larger active sets, please see our response to Reviewer tpr(Q2) for details. Appendix G reports low active-set change rates: 0.30% for SP and 0.14% for KS. The regularity conditions (LICQ, strict complementarity, and H ≻ 0) are standard in convex sensitivity analysis [2,3] and common in differentiable optimization [1,5], thus PEAR does not require stronger assumptions.
>
> ---
> We will revise the manuscript accordingly and include additional results in the appendix.
>
> **Ref.**
>
> [1] Gould et al. (2021), *Deep Declarative Networks*, TPAMI.
>
> [2] Fiacco (1983), *Introduction to Sensitivity and Stability Analysis in Nonlinear Programming*.
>
> [3] Bonnans and Shapiro (2013), *Perturbation Analysis of Optimization Problems*, Springer.
>
> [4] Wilder et al. (2019), *Melding the Data-Decisions Pipeline*, AAAI.
>
> [5] Magoon et al. (2025), *Differentiation through Black-Box Quadratic Programming Solvers*, NeurIPS.
>
> [6] Stellato et al. (2020), *OSQP*, MPC.
>
> [7] Mandi et al. (2025), *Minimizing Surrogate Losses for Decision-Focused Learning Using Differentiable Optimization*, ECAI.

---

> > ### Author Rebuttal · Reviewer_S74w · 2026-04-01
> >
> > Thanks for the explanation. I have no further questions. Best!

---

> > > ### Author Response · Authors · 2026-04-01
> > >
> > > Thank you for the kind words and for confirming that the concerns have been fully resolved. We are grateful for the detailed and constructive feedback, which helped us improve the paper. If the responses and additional results are deemed satisfactory, we would respectfully ask that this be taken into account in your final assessment.

---

### Decision · Program_Chairs · 2026-04-30

**Decision:**

Accept (regular)

**Comment:**

This paper introduces a novel geometric interpretation of regret gradients in Decision-Focused Learning (DFL), demonstrating that the regret gradient can be formulated in closed-form as the tangent-space projection of the prediction error scaled by local curvature. Building on this foundational insight, the authors propose PEAR (Projected Error As Regret-gradient), a method that isolates decision-relevant error components and computes gradients via a reduced linear system over active constraints. By avoiding the computationally expensive process of differentiating through solver iterations, PEAR achieves superior decision quality and computational efficiency across linear and quadratic programming benchmarks, even under constraint shifts. All reviewers find merits in the paper. The author should carefully consider the reviewers' comments to include real-world experiments when preparing the final version.